# Bioavailable phosphite in the surface ocean during the Great Oxidation Event

Abu Saeed Baidya [1] ✉, Joanne S. Boden [1], Yuhao Li [2], Albertus J. B. Smith [3], Kurt O. Konhauser [2,4] & Eva E. Stüeken [1]

Phosphorus availability has influenced the co-evolution of life and Earth's environments. While phosphate was likely the main phosphorus source for life during the Archean, phosphite ($HPO_3^{2-}$) gained importance leading up to the Great Oxidation Event (GOE). However, the concentration of phosphite in seawater at that time, and the processes driving this shift in P utilization, remain poorly constrained. Using lab experiments and phosphite data from banded iron formations (BIFs), we show that hydrous ferric oxides (HFO) preferentially remove phosphate over phosphite. This suggests that shallow seawater at the onset of the GOE could have contained up to 0.17 μM phosphite, comprising 5–88% of total dissolved inorganic phosphorus. We propose that phosphate depletion driven by HFO adsorption and microbial competition may have promoted the use of phosphite as an alternative P source.

Phosphorus is a key element of modern biology, playing a vital role in the formation of phospholipids, cellular energy exchange, and the storage of genomic information in RNA and DNA. Therefore, P must have been crucial to the origin and diversification of life. However, among the major bioessential elements, P is the least abundant element in nature[1]. Its most common geological form, apatite, has low solubility in water, potentially making it the ultimate limiting nutrient in marine ecosystems[2]. Consequently, its availability throughout geologic time has likely affected the trajectory of biological evolution[3,4].

Modern microbial life is heavily dependent on phosphate (P(V)). However, reduced P species, such as phosphonates (molecules with P-C bonds where P is in a 3+ redox state) and inorganic phosphite (P(III)), also serve as important P sources, especially in P(V)-limiting environments[5–8]. They are thus significant in the global P cycle. P(III) is up to 1000 times more soluble than P(V) in the presence of divalent metals[9], suggesting that it may be more bioavailable in marine environments than P(V). P(III) is particularly important to modern microbial life for two reasons. First, it can be utilized as a P source for cellular uptake, a process known as assimilatory phosphite oxidation (APO)[10]. Second, it serves as an electron donor and energy source via dissimilatory phosphite oxidation (DPO)[11]. Although DPO is speculated to

have evolved by 3.3 Ga[10], a subsequent study[6] concluded that P(V) was the main P-source for microbial life at that time. The genes responsible for P(III) metabolism only became more widespread across the tree of life around the Neoarchean-Paleoproterozoic boundary (2.8–2.2 Ga)[6], i.e., just before and during the GOE at around 2.50–2.20 Ga[12–14]. Nonetheless, the processes that might have triggered this shift in P-utilization by microbial life during this time remain unclear.

Both abiotic and biotic processes can produce P(III) in modern environments[15,16]. In the Archean, the proposed abiotic sources of P(III) include; (1) the reduction of P(V) by iron redox chemistry during diagenesis and metamorphism[9,17], (2) serpentinization[18], (3) lightning-induced reduction of P(V)[19,20], and (4) the dissolution of phosphide minerals, such as schreibersite ((Fe,Ni)$_3$P), either delivered by meteorites and/or produced in soils during lightning strikes or in contact metamorphic rocks[21–23]. Iron redox-controlled reduction of P(V), in particular, is facilitated by concomitant oxidation of Fe(II) or by molecular hydrogen ($H_2$) under moderate to high-temperature metamorphic conditions[9,17]. Such processes may happen during serpentinization and high-grade metamorphism in the Archean[9,17]. In addition to the inorganic sources, the major biological source of P(III) is the degradation of phosphonates, which are reduced carbon-phosphorus compounds that

[1]School of Earth and Environmental Sciences, University of St. Andrews, St, Andrews, UK. [2]Department of Earth and Atmospheric Sciences, University of Alberta, Earth Sciences Building, Edmonton, AB, Canada. [3]DSTI-NRF Cimera and PPM Research Group, Department of Geology, University of Johannesburg, Auckland Park, Johannesburg, South Africa. [4]Dept of Earth & Environmental Sciences, University of Manchester, Manchester, UK. ✉e-mail: asb27@st-andrews.ac.uk

life started to metabolize in the Neoarchean[6]. The P(III) produced by these processes might have accumulated in significant amounts in the Archean ocean[9,24], as it is kinetically stable, with a slow breakdown rate in the absence of biological activity (e.g., via DPO or APO), radical ions (e.g., ·OH), and molecular oxygen ($O_2$), having an estimated half-life of 0.6 Ma. Indeed, P(III) has been detected in Eoarchean high-grade metamorphosed BIF and carbonate rocks from the Isua Greenstone Belt and in some recent serpentinites[9,18]. However, data from P(III) in Precambrian sedimentary rocks are limited, and there is also no direct estimation of dissolved inorganic P(III) in Precambrian seawater[6].

Indirect estimates of P(V) concentrations in the Precambrian oceans have been made using P(V) concentrations in rocks including carbonates[25], terrigenous marine sediments[3], and BIFs[26–28]. Among these methods, the approach based on P(V) in BIFs assumes that the latter precipitated as HFO, such as ferrihydrite, in the photic zone overlying the continental shelf[29]. It integrates experimental data on the fractionation of phosphate between HFO and seawater, along with P(V) and Fe concentration in the BIFs[26,28]. The following equation is used to estimate oceanic P(V) concentrations: $[P(V)_d] = (1/K_{ads}) \cdot (P(V)_{ads}/Fe^{3+}_{ads})$; where $[P(V)_d]$ is the concentration (in μM) of dissolved P(V); $P(V)_{ads}$ and $Fe^{3+}_{ads}$ are the concentrations (μM) of adsorbed and precipitated P(V) and Fe on HFO, respectively; and $K_{ads}$ (μM$^{-1}$) is the adsorption coefficient. Typically, $K_{ads}$ (μM$^{-1}$) is experimentally determined, and the $P(V)_{ads}/Fe^{3+}_{ads}$ ratio is directly obtained from BIF analyses. However, this method has never been applied to estimate inorganic P(III) concentrations in the Precambrian ocean. By analogy to P(V), if $K_{ads}$ (μM$^{-1}$) and $P(III)_{ads}/Fe^{3+}_{ads}$ values for P(III) are known, it would be possible to reconstruct the inorganic P(III) concentrations of the Archean-Proterozoic ocean.

We conducted laboratory experiments to simulate the precipitation of BIFs as HFO in various solutions; including deionized (DI) water, 10-times diluted seawater, and seawater (artificially made containing 0.56 M NaCl, 0.055 M Ca, 0.045 M Mg with an ionic strength of 0.86 mol/L), with or without dissolved silica (Si), and varying concentrations of P(V) and P(III) (see Materials and Methods for details). The initial iron concentration was 0.2 mM Fe(II), and dissolved Si concentrations were 0 mM, 0.22 mM, or 2.2 mM, with the highest concentrations reflecting estimated concentrations of the Archean-Paleoproterozic[28,30]. Adsorption tests were performed at a pH of 8 ± 0.2 following previous studies[28,30] as well as at 6.75 ± 0.25 that is consistent with the estimated pH of Archean seawater[31] and for two different experimental durations (0.5 h and 24 h). Adsorption coefficients ($K_{ads}$) were determined in all cases. In addition to the experiments, we also measured concentrations of P-species including P(III), P(V), pyrophosphate (PP(V)), and total P and Fe concentrations in Neoarchean and Paleoproterozoic (2.60–2.46 Ga) BIF samples from five rock formations located in Western Australia (Pilbara Craton) and South Africa (Transvaal Supergroup). We then used the P(III) concentrations of these BIF samples and the experimentally determined $K_{ads}$ of P(III) to estimate the phosphite concentrations in outer shelf settings of the oceans during the GOE. Finally, we explore possible reasons for the shift in microbial P-utilization during the GOE.

## Results and discussion
### Experiments and rock analysis

In our experiments, nearly all (~99%) of the dissolved Fe(II) precipitated as an orange/red phase irrespective of solution chemistry, which is identified as HFO based on XRD and FTIR measurements (Figure S3 in Supplementary Material; Supplementary Dataset 3, 4). The FTIR data of the precipitate in 'Fe-in-seawater-without-Si' experiment show absorption peaks at 3360–3370 and 1641–1645 cm$^{-1}$, which corresponds to OH-stretching and OH-bending, respectively, suggesting that the precipitate has OH in its structure (Figure S3A, B, Supplementary Dataset 3)[32,33]. The XRD data show broad peaks at ~32.4 and 60.5, which are characteristics of ferrihydrite (Figure S3C,

Supplementary Dataset 4)[34,35]. In experiments containing Si, a mixture and amorphous $SiO_2$ and HFO precipitated (Figure S3C). The precipitation of amorphous silica is likely as we used 2.2 mM Si in the experimental solution, which is the saturation limit of amorphous silica in seawater[30]. Previous studies noted that the presence of Si in Fe-Si-$H_2O$ systems like ours may shift the ferrihydrite XRD peaks towards amorphous silica[34,35]. We see the similar effect in P(V) adsorption experimental products (Figure S3D). Despite this shift, the major ferrihydrite peak at ~32.5 is observed in P(III) experimental products (Figure S3C). In summary, HFO precipitated in seawater without Si, while a mixture of amorphous silica and HFO precipitated in seawater containing Si at our experimental conditions.

In contrast to Fe(II), the removal of P-species was variable depending on the specific solution chemistry (Fig. 1,S5; Supplementary Dataset 1). When P-species were removed, they were likely incorporated into the crystal lattices of HFO and adsorbed onto its surface (hereafter referred to as "sorption" to describe the total amount of P(V) or P(III) removed), consistent with previous studies[28,30]. The extent of sorption can be used to calculate $K_{ads}$, with higher values indicating stronger sorption and more effective removal from the solution.

The experiments suggest that salinity has some control on P(V) sorption onto HFO (Fig. 1A). We observed less sorption of P(V) in DI water ($K_{ads}$ = 0.011) compared to 10-fold diluted artificial seawater ($K_{ads}$ = 0.039) (0.22 mM Si; hereafter DiluSeaSi) and concentrated artificial seawater (2.2 mM Si; hereafter SeaSi) (Fig. 1A). Other studies have reported a $K_{ads}$ value of 0.021 for P(V) sorption in artificial seawater with 2.2 mM Si (SeaSi), which has a similar composition as in our experiments[28]. This value is lower than that in DiluSeaSi, suggesting stronger sorption in DiluSeaSi, which may be due to less dissolved Si in this solution. Previous studies also have demonstrated that dissolved Si reduces P(V) sorption in natural seawater and in 0.56 M NaCl, for example $K_{ads}$ values for P(V) in these solutions are 0.338 and 0.064, respectively, while the addition of 2.2 mM of Si reduces the $K_{ads}$ values to 0.008 and 0.002, respectively[28,30]. Despite the presence of dissolved Si in both DiluSeaSi and SeaSi, the markedly lower adsorption coefficient ($K_{ads}$) in DI water indicates: 1) suppressed outer-sphere complexation at low ionic strength[36]; and 2) weakening of P(V) adsorption due the absence of $Ca^{2+}$ and $Mg^{2+}$ cations in DI water to bridge between P(V) and ferrihydrite surface[28].

The experiments further suggest that pH between 6.5 and 8 and experimental duration have limited control on P(V) adsorption (Fig. 1C). The lack of meaningful change in the results between 0.5-hour and 24-hour experiments suggests that our experiments quickly reached equilibrium and remained at a steady state afterwards. More specifically, we note that $K_{ads}$ value for P(V) at pH 6.75 ± 0.25 is 0.026, which is very similar to the $K_{ads}$ value of 0.021 for P(V) at pH 8 as reported by Jones et al.[28]. Furthermore, for 0.5 h and 24 h P(V) adsorption experiments, the $K_{ads}$ changed from 0.026 to 0.027, implying a negligible effect of experimental duration (Fig. 1C). This is also consistent with Jones et al.[28], who also found that the effect of duration for P(V) adsorption at pH 8 is limited.

Importantly, our experiments show that sorption of P(III) onto HFO is very limited, irrespective of solution chemistry (including pH in the range of 7-8) and experimental duration (Fig. 1B, C). The $K_{ads}$ values for P(III) at pH 8 in DI water, DiluSeaSi, and SeaSi are 0.0003, 0.0011, and 0.0008, respectively (Fig. 1B). The $K_{ads}$ value for P(III) at pH 6.75 ± 0.25 in SeaSi is 0.0005, which is indistinguishable from that at pH 8 ± 0.2 ($K_{ads}$ = 0.0008) (Fig. 1B, C). Furthermore, P(III) adsorption after 0.5 h and 24 h in SeaSi at pH 6.75 is indistinguishable, suggesting that equilibrium was reached quickly. We note that all these $K_{ads}$ values are very similar to each other and within error of the experimental procedure indicating a limited or indistinguishable effect of solution salinity, pH, and experimental duration on P(III) sorption onto HFO. Notably, a direct comparison between P(V) and P(III) adsorption coefficients suggests that P(V) is adsorbed more strongly irrespective

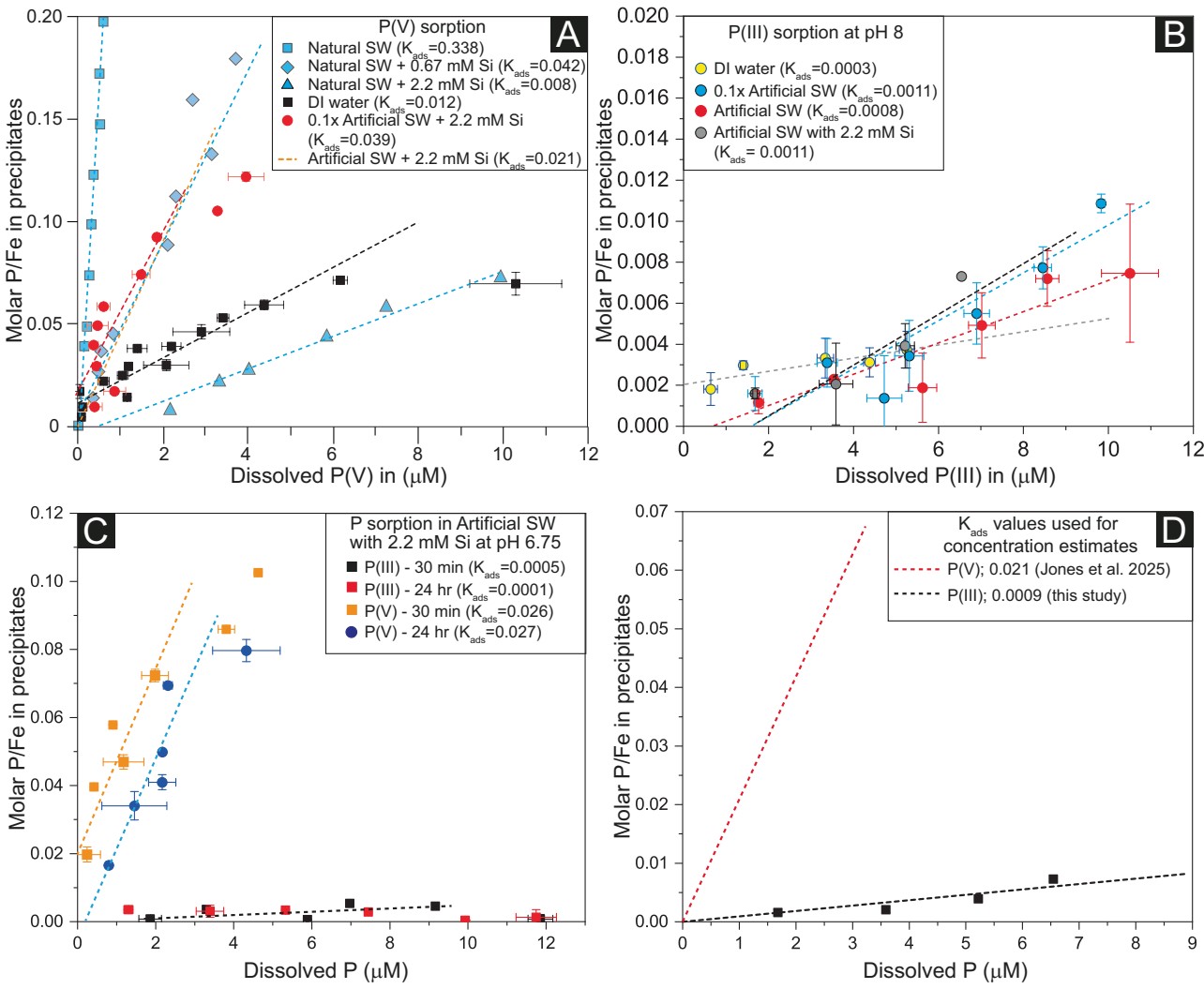

**Fig. 1 | P(V) and P(III) sorption patterns from hydrous ferric oxide co-precipitation experiments. A** compiled P(V) sorption data from previous studies[28] are compared with new data (black and red). SW stands for seawater. 'Natural SW' is the low-nutrient Sargasso seawater whereas 'Artificial SW' is artificially prepared containing NaCl, Ca$^{2+}$, and Mg$^{2+}$ and representing the Archean calcitic seawater. $K_{ads}$ is the coefficient of sorption. Individual datapoints for artificial seawater with 2.2 mM Si were not available, therefore the trend line (orange line) is reconstructed from the $K_{ads}$ value[28]. Data for the natural SW[28] show the effect of Si on sorption while the other three datasets show the effect of salinity. **B**, **C** P(III) adsorption data

generated in this study. P(III) adsorption is limited compared to P(V) irrespective of solution chemistry. Error bars in (**A**–**C**) represent standard deviations of the means. For a direct comparison with P(V) sorption trend lines in (**A**), those in (**B**) are not forced to go through the origin. Two datapoints shown by hollow circles in (**B**) are not considered for producing the trendline because of experimental error. **D** $K_{ads}$ values used for P(V) and P(III) estimates in seawater around the Neoarchean-Paleoproterozoic boundary. As we used the reported $K_{ads}$ value to produce the trend line for P(V), it went through the origin. For a direct comparison, the trendline for P(III) is forced to go through the origin.

of solution chemistry. For example, P(V) sorption is 36, 35, and 26 times stronger than P(III) in DI water, DilSeaSi, and SeaSi, respectively, at pH 8 (Fig. 1A–C). These values imply more efficient removal of P(V) from solution by HFO compared to P(III) under all conditions. The discrepancy of affinity to ferrihydrite is likely due to the incompatibility of the oxygen coordination of phosphite anions with the surface hydroxyl groups on ferrihydrite particles and the lower charge density of P(III) compared to P(V) at the same pH.

Key geological, mineralogical, and compositional features of the studied BIF are summarized in Table 1. These rocks formed between 2.60 and 2.44 Ga and experienced burial metamorphism at temperatures ranging from 110 to 170 °C for the South African BIF[37] and 160–360 °C for the Western Australian BIF[38] (further details are avail in the Materials and Methods section). XRD data reveals variable proportions of quartz, magnetite, hematite, siderite, ankerite, with minor amounts of pyrite, riebeckite, and stilponomelane (Figure S2, Supplementary Dataset 4). The Fe contents of the samples range from 27

to > 50 wt.%. The Joffre Member has comparatively low total P ranging from 20 to 110 ppm, while the other four formations contain higher total P levels, ranging from 20 to 3720 ppm (Fig. 2).

Figure 2 summarizes the P speciation data from the EDTA-NaOH extracts and total P contents (Supplementary Dataset 2). The EDTA-NaOH solution extracted only a small portion (1–38%) of total amount of P present in the solid samples; however, these yields are consistent with those reported in previous studies[9]. The associated uncertainties in seawater P(III) reconstructions are discussed below. All the studied samples contain P(III), with the highest concentrations found in the Kuruman Iron Formation and the lowest in the Joffre Member. The concentration of P(III) is consistently lower than that of P(V) in the extraction solutions.

## Estimation of P(III) in seawater around the GOE
The $K_{ads}$ values obtained from our sorption experiments, combined with P-speciation data in BIFs, allow us to estimate P(III) and P(V)

**Table 1 | Geological and chemical features of the banded iron formation samples**

| Location | Age | Max. Meta. Temp | No. of Samples | Mineralogy (XRD) | P(III) (ppm)* | P(V) (ppm)* | Total P (ppm)* | Total Fe (wt%)* | Extraction yield (%) |
|---|---|---|---|---|---|---|---|---|---|
| Kuruman -Gamohaan Iron Formation[$] | 2.55–2.44 Ga | 170 °C | 7 | qtz, mag, hem, sid, anke, stilp(?), py(?) | 0.22–0.37 | 1.59–28.4 | 70–950 | 26.8–44.8 | 1.96–4.53 |
| Joffre Base Member[£] | 2.46 Ga | 200–360 °C | 4 | qtz, mag, hem, sid, anke, py, riebe, stilp(?) | 0.02–0.05 | 2.21–15.4 | 20–110 | 27.4–34.7 | 11.3–37.6 |
| Dales Gorge Member[£] | 2.49–2.46 Ga | 200–360 °C | 6 | qtz, mag, hem, anke, py, riebe, stilp(?) | 0.15–0.56 | 1.59–60.6 | 20–3720 | 29.6–> 50 | 1.65–12.5 |
| Marra Mamba Formation[£] | 2.6 Ga | 200–360 °C | 4 | qtz, mag, sid, anke, py, stilp(?) | 0.23–0.34 | 3.32–11.6 | 160–1090 | 30.4–> 50 | 0.95–2.34 |

* P(III) and P(V) concentration in EDTA-NaOH extractable rock; # concentration in bulk rock; Minerals with '?' mark are possibly present. $ South Africa, £ Western Australia, *qtz* quartz, *mag* magnetite, *hem* hematite, *sid* siderite, *anke* ankerite, *stilp* stilpnomelane, *py* pyrite, *riebe* riebeckite.

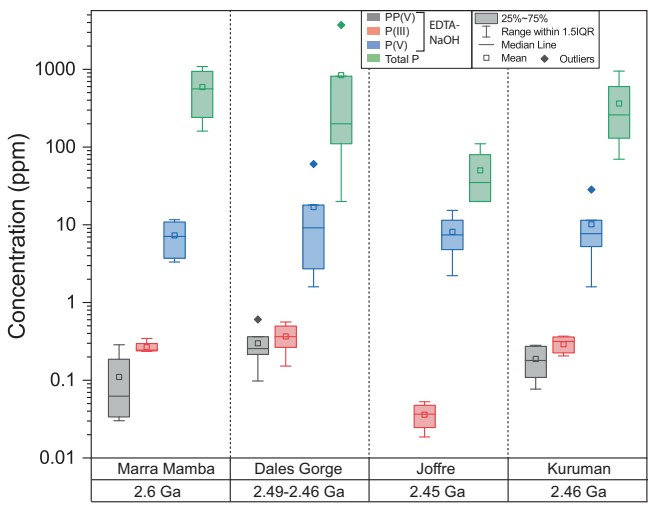

**Fig. 2 | Box plots show the P speciation data in the banded iron formation rocks.** Detectable amounts of P(III) are present in all the EDTA-NaOH extracts of the banded iron formation samples and its concentration is lower than P(V). Total extracted amount of P-species (P(III), P(V), and pyrophosphate (PP(V)) are lower than total P in the solid samples.

concentrations in seawater around the Neoarchean-Paleoproterozoic boundary. This estimation relies on several assumptions: First, we presume that the studied BIFs precipitated as HFO. The precipitation of BIF is debated, with proposed precursors including HFO[39], greenalite[40], green rust[41], magnetite[42], and siderite[43]. Among these, HFO is the most significant globally[39], particularly for BIF containing magnetite and hematite, as is the case in this study (Fig. S3). If the primary mineral was instead composed of Fe(II)[40], the observed Fe(III) phases including magnetite and hematite would have to be of secondary origin. However, the complete absence of primary Fe(III) is unlikely, given hydrological constraints[44], independent evidence of oxic conditions in Neoarchean surface waters[45,46], highly fractionated Fe isotopes in these BIFs indicating depositional and post-depositional redox cycling[47], and the incontestable fact that a biosphere capable of oxidizing dissolved Fe(II) existed at that time[29]. HFO, the most likely precursor of preserved Fe(III) minerals in BIFs, formed in the photic zone of the water column overlying the continental shelf (i.e., top 100 meters)[39], while Fe(II) mineral precipitation was dominant in the deep ocean, from proximal to hydrothermal Fe-sources[48] to the continental slope before upwelling onto the shelf[49]. In shallow water settings overlying the shelf, either free $O_2$ or photoferrotrophs or both could have facilitated the Fe(II) oxidation and HFO precipitation. Therefore, our estimation is valid for shallow-water settings where HFO precipitated. Our results are not relevant for deep-water settings where more reduced Fe(II) phases such as greenalite likely precipitated. We also emphasize that the shallow photic zone is most relevant for primary producers, and therefore constraints on the P-inventory of this habitat are critical.

Second, the amount of P(III) detected in the EDTA-NaOH extraction cannot be used directly to calculate the total sorbed P(III) during BIF precipitation without considering two issues: (1) the preferential extraction of P(III) over P(V) by the EDTA-NaOH solution, due to the former's higher solubility[9], and (2) the post-depositional transformation of P(V) into P(III) facilitated by iron redox chemistry[9,17].

Previous studies have reported low extraction yields (< 3%) of P species in EDTA-NaOH solutions from solid rocks[9]. To the best of our knowledge, no study to date has specifically examined whether P(III) can be preferentially leached. Therefore, we have considered two extreme possibilities: (1) the ratio of P species in BIF is the same as in the EDTA-NaOH extract, and (2) complete extraction of BIF-hosted

P(III) into EDTA-NaOH solution. These scenarios help bracket the potential P(III) concentrations in the BIF. To address the post-depositional transformation of P(V) into P(III), we considered three possible cases: (1) no metamorphic P(III), meaning that all measured P(III) represents primary sorbed inorganic P(III) during BIF precipitation; (2) a mixture of sorbed and metamorphic phosphite; and (3) all the P(III) is metamorphic in origin. In case (3), it is impossible to estimate the original P(III) concentration from BIF precipitation, as it implies that none of the detected P(III) was originally sorbed to BIF.

Together, these permutations lead to five scenarios to translate the measured P(III) and P(V) in the BIFs into seawater P(III) and P(V) concentrations (Table 2). We note that Scenario 5, where all P(III) is metamorphic in origin, is excluded from the compilation as here the calculated seawater value would be zero. Scenarios 2 and 4 are further subdivided, depending on the metamorphic constraint on P(V) reduction into P(III) in ferruginous diagenetic and metamorphic environments[9,17]. The highest estimated values for P(V) range from 0.01 to 0.93 μM with minor variations across the different scenarios (Table 2, Fig. 3). These estimates are lower than estimates based on carbonates[25], hydrothermal vent precipitates[48,50], as well as experiments and modelling[51] but similar to predictions from several other geochemical estimates based on BIF samples[26,28] and genomic estimates[6] (Fig. 3).

To estimate post-depositional metamorphic P(III), we used published experimental yields of metamorphic and diagenetic P(V) reduction in ferruginous conditions[9,17]. Baidya et al.[17] conducted several experiments at 350 °C, which is close to the highest metamorphic temperature experienced by the studied BIF[38], and reported a yield of 0.075%. They also demonstrated that magnetite inhibits the reduction of P(V) to P(III) even in the presence of $H_2$ at 350 °C[17]. Given that magnetite is consistently present in the BIF samples (Table 1), metamorphic P(III) may be limited, making Scenario 4a and Scenario 5 - where all BIF-bound P(III) is metamorphic - less plausible. Among the remaining scenarios, Scenario 1, which assumes the same ratio of P-species in BIF samples as in the EDTA-NaOH extract and no additional P(III) formation during diagenesis and metamorphism, provides the highest possible concentrations of P(III) in seawater, ranging from 1 to 165 nM (Table 2). Importantly, the estimated P(III) and P(V) concentrations in Scenario 1 suggest that P(III) could have constituted 5-88% of total dissolved inorganic P (P(V) + P(III)) in seawater at the onset of the GOE (Fig. 3).

## Shift in microbial P-utilization around the GOE

Phylogenetic studies suggest that microbial communities began utilizing P(III) between 2.6–2.2 Ga[6]. If this is the case, there must have been sufficient P(III) in seawater to facilitate this evolutionary shift. So far, there are limited data on P(III) concentrations in the modern ocean and its relation to microbial growth. P(III) was not detected in the tropical Atlantic Ocean[15] but has been reported in geothermal pools (0.06 ± 0.02 μM)[16], lakes (0.01–0.71 μM)[52,53], rivers (0.08–0.9 μM)[54], and ponds (0.14–2.90 μM)[54]. Experimental studies have found that P(III)-dependent microbial growth is possible at 50 μM P(III)[7]. Our estimated concentrations of P(III) (0.00–0.17 μM) in shallow seawater at the onset of the GOE coincide with the lower end of the observed P(III) range in above-mentioned natural environments, some of which (for example geothermal pools and surface oceans) are inhabited by P(III)-dependent microbial life[7,55]. We therefore suggest that our estimated concentration of phosphite might have been sufficient for the growth of microbial life during the GOE.

Sources and sinks of P(III) during the GOE are poorly constrained and they may depend on several parameters including the spatial distribution of P(III) in ocean (i.e., deep ocean vs. shallow ocean) and time. As noted above, major inorganic sources of P(III) in the Archean may include meteoritic delivery[22], lightning-induced formation of phosphides[19,20], metamorphism of ferruginous sediments[9,17], and

**Table 2 | Estimates of phosphite and phosphate concentrations (μM) in surface ocean**

| Locations | Scenario 1<br>1. P species ratio in BIF is same as in EDTA-NaOH extract<br>2. Metamorphic phosphite is none | | Scenario 2a/2b<br>1. P species ratio in BIF is same as in EDTA-NaOH extract<br>2. Metamorphic phosphite using exp. yield | | Scenario 3<br>1. 100% phosphite is extracted from BIF<br>2. Metamorphic phosphite is none | | Scenario 4a/4b<br>1. 100% phosphite is extracted from BIF<br>2. Metamorphic phosphite using exp. yield | |
|---|---|---|---|---|---|---|---|---|
| | P(III) | P(V) | P(III) | P(V) | P(III) | P(V) | P(III) | P(V) |
| Kuruman-Gamohaan | 0.024–0.075 | 0.013–0.352 | 0.000*–0.054[a]<br>0.024–0.075[b] | 0.014–0.354[a]<br>0.013–0.352[b] | 0.001–0.002 | 0.016–0.354 | ENP[a]<br>0.001–0.002[b] | 0.016–0.354[a]<br>0.016–0.354[b] |
| Joffre | 0.000*–0.002 | 0.005–0.029 | 0.000*[a]<br>0.000*–0.001[b] | 0.005–0.029[a]<br>0.005–0.029[b] | 0.000* | 0.005–0.012 | ENP[a]<br>0.000*[b] | 0.005–0.012[a]<br>0.005–0.029[b] |
| Dales Gorge | 0.017–0.142 | 0.002–0.921 | 0.000*–0.020[a]<br>0.017–0.142[b] | 0.003–0.927[a]<br>0.002–0.921[b] | 0.001–0.002 | 0.004–0.927 | ENP[a]<br>0.001*–0.002[b] | 0.003–0.927[a]<br>0.003–0.927[b] |
| Marra Mamba | 0.058–0.165 | 0.024–0.300 | 0.000*–0.033[a]<br>0.058–0.165[b] | 0.025–0.307[a]<br>0.024–0.300[b] | 0.001–0.002 | 0.027–0.307 | ENP[a]<br>0.001–0.002[b] | 0.027–0.307[a]<br>0.027–0.307[b] |

[a]Scenario 2a/4a- Experimental yield of Herschy et al.[9]; [b]Scenario 2b/4b- Experimental yield of Baidya et al.[17]; exp experimental, BIF banded iron formation, ENP estimation not possible (all phosphite are metamorphic like Scenario 5); *: values 0.000 means the concentration is less than 0.5 nM.

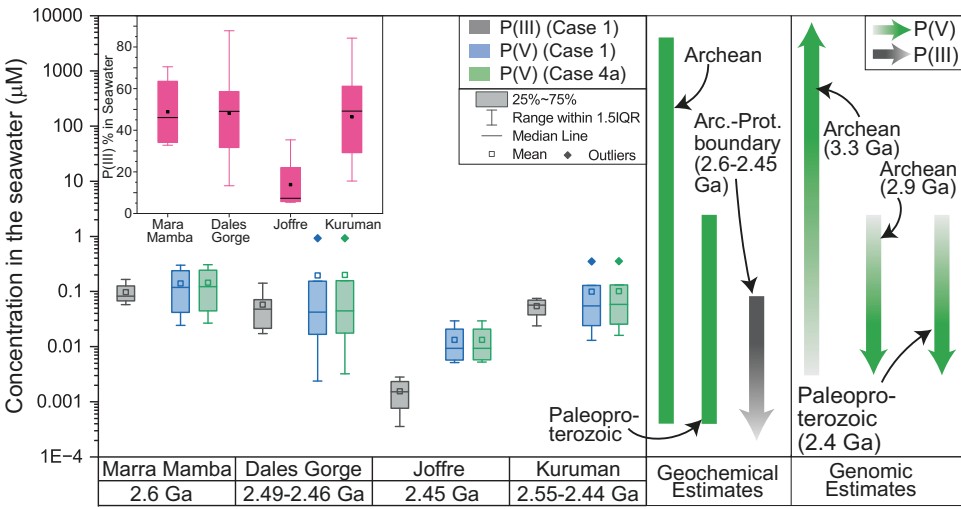

**Fig. 3 | Estimated P(III) and P(V) concentrations in ocean at the onset of the Great Oxidation Event.** 'Arc.' and 'Prot.' stand for 'Archean' and 'Proterozoic', respectively. The boxplots show the estimated concentrations of P(III) and P(V) in two extreme scenarios, (1) and (5), which provide the highest and lowest possible concentrations of P(III), respectively. In scenario 5, all the P(III) assumed to be metamorphic implying P(III) estimation in Precambrian seawater is not possible.

The inset box plot shows the P(III) proportions of total dissolved inorganic P (P(III) + P(V)) in scenario 1. The green lines and arrows show the estimated P(V) concentrations in the Archean and around the Neoarchean-Paleoproterozoic boundary by previous geochemical[3,25,50,51,59,81] and genomic approaches[6]. The grey arrow is the estimation of P(III) in this study.

serpentinization[18]. During the GOE, meteoritic delivery might have been limited but the other abiotic sources could have been active. As our estimated concentration is more relevant for relatively shallow waters, *i.e.* outer-shelf settings where BIFs were forming, deep-sea sources such as serpentinization might have contributed less unless P(III) from serpentinization was transported to shallow waters via upwelling and long-range oceanic transport in the form of hydrothermal plumes[56]. However, the relative contributions of these sources are unknown. Importantly, biological phosphite production could have started during the earliest stages of the GOE around 2.5 Ga. Although no direct biological conversion of P(V) compounds to P(III) is known, biological production of phosphonate from P(V) sources is common in the modern ocean[15], and P(III) may form via disintegration of phosphonates[54]. A recent phylogenetic study demonstrated that organisms developed the ability to generate phosphonate during the GOE[6]. Therefore, phosphite production due to phosphonate degradation could have been possible during the GOE. Hence, both abiotic and biotic processes could have contributed phosphite at the onset and during the GOE. Regarding phosphite sinks, one major sink could have been microbial usage during the GOE, particularly in the shallow ocean where productivity was perhaps the highest. In addition, P(III) incorporation into iron oxides would have constituted a minor sink, as indicated by our data. Other inorganic sinks of P(III) remain to be explored. However, irrespective of these uncertainties about P(III) source and sink fluxes, our data suggest that there was a standing reservoir of P(III) in seawater in the late Archean and early Proterozoic that has not previously been recognized.

Understanding whether P(III) was used by microbial life for assimilatory phosphite oxidation (APO) or dissimilatory phosphite oxidation (DPO) at the onset of GOE is crucial to determine if P(III) was used primarily as a P-source or for energy gain. Experimental studies imply that higher P(III) concentrations are required for DPO compared to APO (0.1–10 μM P(III) for APO and ≥10 μM for DPO)[7,10,11,57]. We analysed a previously published phylogenetic tree of *ptxD*[6], which emerged between 2.3 and 2.2 Ga[6], and is used in both APO and DPO (see Materials and Method section for further details). The tree reveals that homologs from bacteria performing DPO form a monophyletic group with a posterior probability of 100, suggesting that DPO evolved

once and subsequently radiated into different species. When rooted with minimal ancestor deviation from Tria et al.[58], the most parsimonious explanation for the evolution of *ptxD*s is that the earliest *ptxD*s from 2.3 to 2.2 Ga were associated with APO, and DPO-associated homologs evolved later (see Figure S1 in Supplementary Material). In the alternative scenario where the first *ptxD*s were used for DPO, two switches from DPO-associated *ptxD* to APO-associated *ptxD* would be required, which is less parsimonious – and hence less likely – than the one switch required if the first *ptxD* was associated with APO. Furthermore, a different gene, *ptxB*, which imports P(III) for APO, evolved between ~2.6 and 2.3 Ga, pre-dating the *ptxD*s. Consequently, we suggest that microbes were utilizing P(III) for APO by the onset of the GOE, i.e. as a source of P. The low estimated P(III) concentration in seawater at that time (Fig. 3) supports this explanation.

We posit that increased primary productivity and the preferential removal of P(V) compared to P(III) due to BIF precipitation created P(V)-depleted environments in the surface ocean. Major BIFs were precipitated on the outer continental shelves between 2.65 and 2.40 Ga[39], just prior to, and concomitant with, the GOE (Fig. 4C). Dissolved Si is known to reduce the effect of P(V) sorption onto HFO during BIF precipitation, while dissolved Ca and Mg may mitigate the effect of Si[28,30], leading to variable removal of P(V) from seawater during BIF precipitation depending on Si, $Ca^{2+}$, and $Mg^{2+}$ concentrations. Nevertheless, the presence of P in BIF samples worldwide[26] suggests that BIF precipitation indeed removed a portion of the dissolved P(V) from the surface ocean at the onset of GOE. Furthermore, it is generally believed that primary productivity was limited in the Archean due to a range of factors, including lower availability of electron donors (e.g., $Fe^{2+}$ and $H_2$) necessary for anoxygenic photosynthesis (versus water used by oxygenic photosynthesis)[59,60], the absence of exoenzymes to breakdown complex organic macromolecules into useable products for anaerobic respiration[61], less emergent continental landmass and thus less habitable space for microbial mats[62], and higher UV radiation due to the absence of an ozone layer[63]. With the expansion of oxygenic photosynthesis, primary productivity might have increased ten-fold compared to early Archean times[60] (Fig. 4A). Such an extreme increase in biological productivity likely depleted the surface oceans in nutrient elements, particularly

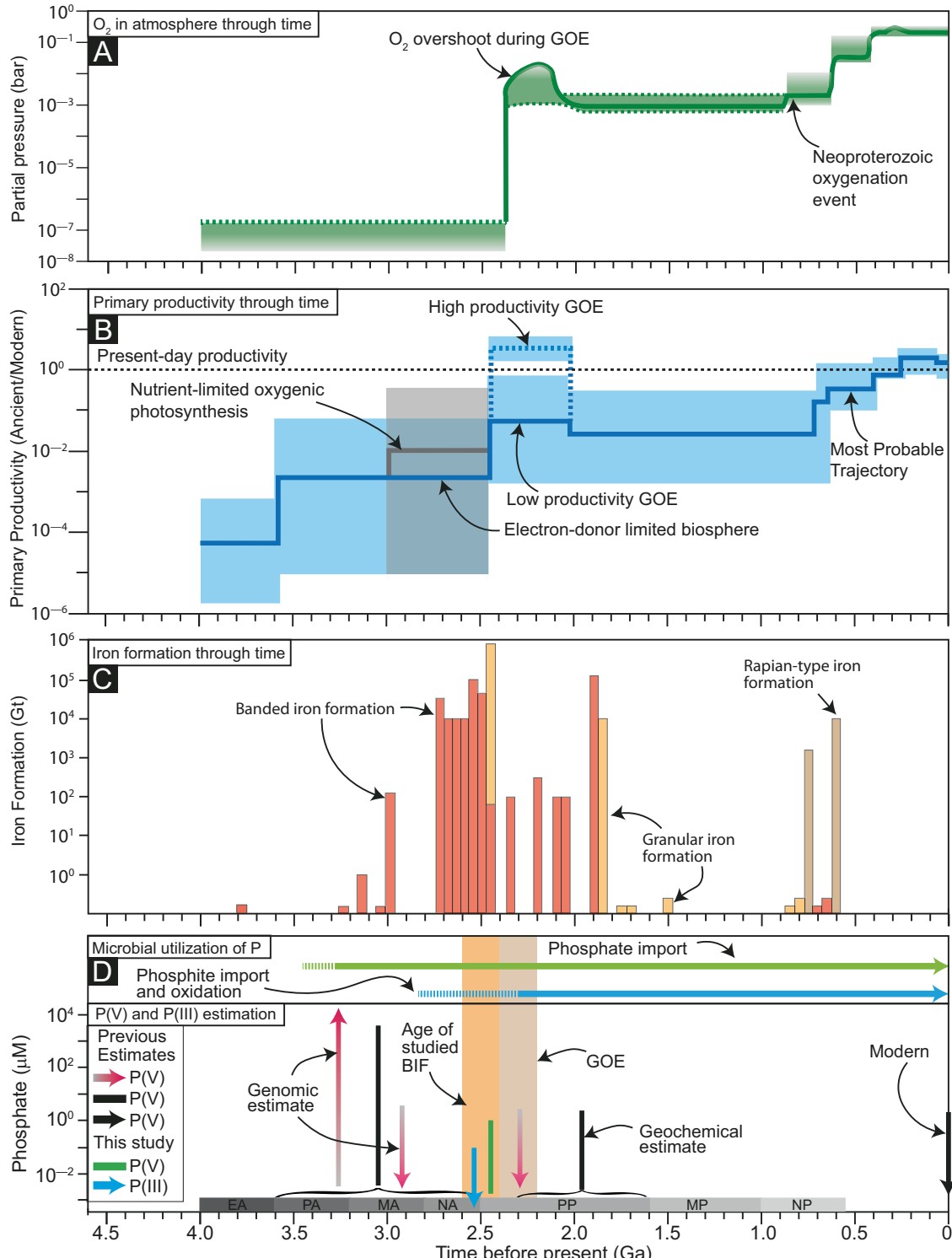

**Fig. 4 | Temporal evolution of key parameters related to microbial P-utilization in deep time.** GOE and BIF stand for Great Oxygenation Event and banded iron formation, respectively. **A**, **B** show the temporal evolution of $O_2$ in the atmosphere and primary productivity in the ocean, respectively[82,83]. Primary productivity increased around the GOE. **C** shows the amount of precipitated BIF, reaching a maximum between 2.65 and 2.40 Ga[39]. **D** shows the timing of microbial P utilization along with known estimates of P(V) in the Archean and Paleoproterozoic[6].

P(V). Therefore, during the GOE, there might have been some locales in the surface oceans with very limited P(V) due to a combined effect of sorption onto HFO and increased primary productivity.

Limited availability of P(V) in parts of the surface ocean aligns with several geochemical and genomic estimates of P(V) in Precambrian

seawater and the data on microbial P(III) utilization. First, we note that higher concentrations of dissolved P(V) (1–4000 μM) in the early Archean oceans are suggested by recent genomic[6] and several geochemical estimates[25,48,51]. We further note that there is a difference in the P(V) abundance in shallow and deep water in the Archean, as

previous studies suggested that the deep water P(V) content in the Archean was 5-65 times higher than that in the modern ocean (2.3 μM), based on jaspilite and hydrothermal vent deposits[48,50], while the surface water was comparatively depleted in P(V) (4–12 times higher than in the modern ocean), based on shelf carbonate chemistry[25]. During the GOE, the P(V) concentration in the surface ocean appears to have declined to submicromolar levels as indicated by both geochemical and genomic data[3,6]. Second, our P(III)/P(V) data suggest that, on average, more than 45% (reaching up to 88%) of total dissolved P in the surface ocean was P(III) during deposition of the Marra Mamba, Dales Gorge, and the Kuruman iron formations before the onset of GOE, further pointing to P(V) depletion in surface ocean (Fig. 3). Third, our genomic analysis suggests that assimilatory phosphite oxidation or APO evolved at this time, a pathway by which microorganisms convert P(III) into P(V) before utilizing it. Such a pathway is not likely to be required if P(V) is abundant in the first place. For example, in the modern surface oceans, organisms have more phosphonate assimilation genes when there is a scarcity of P(V)[5]. These data collectively suggest that there were at least some locales in the surface ocean with P(V) deficiency at the onset of GOE. We further note that at this time, genes responsible for metabolizing reduced P species including P(III) did evolve[6]. In modern P(V)-depleted environments, microbial life uses alternative reduced P species such as phosphonate[6,7]. Therefore, we hypothesize that the P(V) deficiency in surface oceans at the onset of GOE may have facilitated the evolution of genes responsible for utilizing alternative P sources including phosphite, which could have remained in the ocean water even after BIF precipitation due to its lower sorption affinity onto HFO. These reduced P species may have supported productivity above the levels that were sustainable by the shrinking P(V) reservoir and thus played an important role in supporting the rate of biological $O_2$ production that ultimately triggered the transition from an anoxic to an oxic world.

In summary, we estimate the maximum dissolved P(III) concentrations (1–165 nM) in surface ocean around the time of the GOE, which could have constituted 5–88% of the total dissolved inorganic P (P(V) + P(III)) at that time and may thus have helped sustain primary productivity during this crucial time in Earth's history. The stark contrast between P(V) and P(III) sorption on HFO identified by our experiments, as well as the observed prevalence of P(III) compounds in P(V)-depleted settings in the modern ocean, uncovers a potential linkage between the expansion of oxygenated surface waters, increase in primary productivity facilitated by oxygenic photosynthesis, the accumulation of iron oxide minerals on continental shelves, and the radiation of novel P-metabolisms across the tree of life. Our findings thus reveal a previously unknown factor contributing to the co-evolution of Earth and its biosphere.

## Methods

### Adsorption experiments

The laboratory experiments simulated the co-precipitation of banded iron formations (BIFs) as hydrous ferric oxyhydroxides (HFO) and inorganic phosphate or phosphite in deionized water, 10-times diluted seawater, and seawater. Acid-washed (1–2 M HCl) and baked (500 °C) glass containers, acid- and hot water-washed centrifuge tubes, syringe, and pipette tips were used during all stages of the experiments, subsequent sampling, and analysis. $FeCl_2.4H_2O$ (Sigma Aldrich), $NaH_2PO_4$ (Thermo Fisher), $NaH_2PO_3·5H_2O$ (Thermo Fisher), and $Na_2SiO_3$ (Thermo Fisher) were dissolved in deionized water for preparing stock solutions of 20 mM $Fe^{2+}$, 1 mM phosphate and phosphite, and 22 mM Si. The $Fe^{2+}$ solution was freshly prepared before each set of experiments to avoid significant oxidation under the present atmosphere. Artificial seawater containing 0.56 M NaCl (Sigma Aldrich), 55 mM $Ca^{2+}$ ($CaCl_2$, Thermo Fisher), and 45 mM $Mg^{2+}$ ($MgCl_2.6H_2O$, Thermo Fisher) with an ionic strength of 0.86 mol/L was prepared by dissolving the salts in deionized water. This composition represents Precambrian Si-

bearing calcitic sea[28]. Stock solutions were diluted to produce 10 ml experimental solutions containing 0.2 mM $Fe^{2+}$ and 0–28 μM phosphate or phosphite with or without Si of 0.22 mM (10x-diluted seawater) or 2.2 mM (seawater). As an example, to prepare a 2 μM phosphite-bearing artificial sweater solution, we mixed 200 μL of 100 μM phosphite, 100 μL of 20 mM $Fe^{2+}$, and 9.7 mL artificial seawater. The experimental solutions were then mixed with diluted NaOH (variable combinations of 0.01 M, 0.025 M, and 0.05 M) to make an alkaline pH that helps to oxidize $Fe^{2+}$ in the presence of atmospheric oxygen and precipitate HFO. In the case of 2.2 mM Si-bearing solutions, the pH was already alkaline, therefore, addition of NaOH was either not required or was small in volume. 2–5 min time were required for the initial stabilization of pH. Thereafter, a constant pH of 8 ± 0.2 or 6.75 ± 0.25 was maintained for half an hour and adjusted with diluted NaOH and HCl (variable combinations of 0.0.01 M, 0.025 M, and 0.05 M). For the 24 h experiments, the pH was monitored after 0.5, 1, and 20 h and adjusted accordingly. We kept track of the total amount of NaOH and HCl added to each solution to accurately determine the dilution factors at the end of the experiments. The pH was monitored using a pH probe (Hanna Instruments), which was calibrated before every set of experiments. All experiments were performed as doublets or triplets. After the experiments, the solutions were filtered with previously washed (10 ml deionized water) 0.2 μm PTFE hydrophilic (Fisher) filters. We discarded the first 3 ml of the solution after filtering to avoid any contamination from the filter and collected 1 ml, which was immediately acidified with 2% ultrapure $HNO_3$.

All the experimental solutions were diluted 10–100 times with 2% $HNO_3$, and the concentrations of $^{31}P$ and $^{56}Fe$ were measured with a Thermo Scientific Element 2 high resolution inductively coupled plasma mass spectrometer (ICP-MS) equipped with an auto sampler (Elemental Scientific Inc.), a 0.1 ml/min nebulizer, and a Scott spray chamber. Standards containing 0.01 μM to 2.5 μM of Fe and phosphate or phosphite were prepared by dissolving $FeCl_3$ and $NaH_2PO_4$ or $NaH_2PO_3·5H_2O$ (Thermo Fisher, same salt used for experiments) in the same saline matrix as that of the samples. The ICP-MS was operated at a sample gas flow rate of 1 ml/min, cool gas flowrate of 16 ml/min, and RF power of 1250. The $^{31}P$ and $^{56}Fe$ intensities of the sample solutions were measured in medium resolution mode, and concentrations were calculated offline with respect to the standards. Each sample and standard were measured twice, and the average intensities were used for assessment. A few standards were measured at the beginning of the ICP-MS sequence as well as in the middle and at the end to quantify the drift of the ICP-MS and corrections were made when it was required.

### Location and geology of the banded iron formation samples

The Marra Mamba Iron Formation samples were collected from drill core WRL-1, while the Dales Gorge Member samples were collected from drill core DGM-1. Both were provided by the Perth Core Library of the Geological Survey of Western Australia. The Joffre Member samples were obtained from core sample DD98SGP001 via the Rio Tinto core library in Perth. The Gamohaan and Kuruman Formation samples were recovered from drill core DI1, originally drilled and stored by a mining company (Gefco) at Derby, approximately half way between the towns of Kuruman and Danielskuil in the Northern Cape Province of South Africa[64].The sample from the Gamohaan Formation was taken from the uppermost Tsineng Member[65], whereas the samples from the Kuruman Formation were taken from four different members throughout its stratigraphy.

The Hamersley Group comprises about 2.5 Km of consecutive sedimentary and volcanic rocks located within the ca. 80,000 km[2] Hamersley Province of the Pilbara craton in Western Australia. It comprises five IF units, in ascending order the 2.60 Ga Marra Mamba Iron Formation, the 2.48 Ga Dales Gorge Member of the Bockman Iron Formation, the 2.46 Ga Joffre Member of the Brockman Iron Formation, the 2.45 Ga Weeli Wolli Formation and the uppermost Bolgeeda

Iron Formation which is approximated at 2.44 Ga[66]. Metamorphic grade for the Hamersley Basin units have been interpreted from the widespread presence of the minerals prehenite, pumpellyite, epidote and actinolite, which corresponds to a maximum temperature range between 200 and 360 °C[38,67].

The Gamohaan Formation is approximately 110 m thick and is the uppermost formation of the approximately 1600 m thick Campbell-rand Subgroup of the lower Ghaap Group of the Griqualand West region of the Transvaal Supergroup of southern Africa[65]. Although the Campbellrand Subgroup is dominated by stromatolitic dolostone and limestone, the Gamohaan Formation contains a BIF (originally described as an Fe-rich banded chert), called the Tsineng Member, at its top[65] that was sampled for this study. The depositional age range for the upper Campbellrand Subgroup is approximately 2.55 to 2.52 Ga[68]. The Kuruman Formation is the lower formation of the Asbesheuwels Subgroup, which directly overlies the Campbellrand Subgroup[64,69]. Together with the overlying Griquatown Formation, the Asbesheuwels Subgroup comprises 385 to 1000 m of continuous micritic and granular iron formation[64]. The depositional age range for the Kuruman Formation is approximately 2.48 to 2.44 Ga[68]. Together with the correlative Penge Formation in the Transvaal region[68], it is the oldest iron formation of the Transvaal Supergroup. Other than the Griquatown Formation, the Transvaal Supergroup in the Griqualand West region contains two more iron formations in the approximately 2.43 Ga Koegas Subgroup[70] (Schröder et al. 2011) and four iron formations interbedded with manganese beds in the approximately 2.41 Ga Hotazel Formation[71,72]. Estimated burial temperatures of the Kuruman Formation is 100–150 °C[73]. A similar burial temperature is inferred for the Gamohaan Formation as it directly underlies the Kuruman Formation.

### Solid characterization using powder X-ray diffraction (PXRD)
The powdered rock samples and the experimental precipitates were loaded into 0.5 mm or 0.7 mm capillary tubes and sealed for XRD analysis. The PXRD patterns were recorded on a STOE STADIP diffractometer using Mo Kα1 radiation at room temperature from 2.5° to 37° (2θ) with a scan rate of 2.5–3.0° (2θ)/step in capillary Debye-Scherrer mode. The PXRD data were compared to solids in the Inorganic Crystal Structure Database (ICSD) for phase identification using the Crystal Diffract software (version 6.9.3).

### Fourier transform infrared spectroscopy (FTIR)
Fourier transform infrared spectroscopy (FTIR) analysis of experimental precipitates was performed using a Shimadzu spectrometer with a resolution of 4 cm$^{-1}$ and a scanning frequency of 32 min$^{-1}$ at room temperature. Spectra were recorded in the 4000 – 400 cm$^{-1}$ region.

### Whole rock analysis
Approximately 0.30–0.60 g of powder from each of the samples was sent to Australian Laboratory Services (ALS) in Dublin, Ireland, for whole-rock geochemical characterisation using their method ME-MS-61r of four-acid digestion (HCl, HNO$_3$, HF, HClO$_4$) followed by ICP-MS and -AES analyses. Reproducibility was assessed with rock standards OREAS-45d, OREAS-905 and MRGeo-08, and with sample replicates. It was found to be 5 % or better for P and Fe.

### Quantification of P species in the banded iron formation samples
An aliquot (ca. 0.2–0.25 gm) of the powdered samples was treated with an Ethylenediaminetetraacetic acid-sodium hydroxide (0.05 M EDTA and 0.25 M NaOH) solution[18] maintaining a solid:solution ratio of 1:10 for 14–15 h. Na$_2$EDTA (Sigma Aldrich) salt and 10 M NaOH solution (Thermo Scientific) were dissolved in deionized water to make the EDTA-NaOH solution mixture. Acid- and hot-water washed 10 ml

Falcon tubes were used during the extraction procedure. The solutions were then centrifuged at 3000 rpm for 15–20 min. In most cases, the solution was transparent after centrifuging, suggesting the precipitation of all the extracted Fe. In a few cases, the solution was yellow to orange, which suggested the presence of dissolved Fe. Such solutions were further treated with 1 M NaOH to precipitate all the Fe, which is essential for the P speciation measurements using the subsequent Ion Chromatograph (IC)-ICPMS analysis[74]. This is because excess dissolved iron may precipitate as oxides in the anion separation column of the IC and bind phosphate by adsorption within the column, thereby impacting analytical quality.

Four phosphorus species, namely hypophosphite, phosphite, phosphate, and pyrophosphate were analyzed using the IC-ICPMS set-up of Baidya and Stüeken[74]. In this IC-ICPMS set-up, a Thermo Scientific Dionex ICS-6000 IC equipped with a Dionex AS-AP autosampler, a 25 mm Dionex IonPac AS17-C separation column (2 mm bore), a 25 mm Dionex IonPac AG17-G guard column (2 mm bore), and a Dionex ADRS-600 (2 mm) suppressor were used to separate the P species in the solution. The flow rate in the IC was held constant at 0.5 ml/min while the concentration of the KOH eluent solution was ramped up from 1 mM to 40 mM over 20 min. This maximum KOH concentration was held constant for another 22 min followed by a ramp down to 1 mM over 8 min. The suppressor outlet of the IC was physically connected to a 1 ml/min nebulizer attached to the spray chamber (Scott model; quartz glass) of the Element 2 ICP-MS. The IC-PMS was operated at a sample gas flow rate of 1.1 ml/min, cool gas flowrate of 16 ml/min, and RF power of 1183 in medium resolution mode. Data were collected in the ICP-MS as chromatographs of 3 min duration (one minute for monitoring the pre-peak background, one minute for the peak, and one minute for monitoring post-peak background) for each P-species. The chromatographic data were smoothened with the OriginLab software, using the fast furrier transform filter with a points-of-window value of 5, and the peak area under the curve was used for quantification of phosphorus. Standards of the four P species (prepared from NaH$_2$PO$_2$·H$_2$O (Thermo Fisher), Na$_2$HPO$_3$·5H$_2$O (Thermo Fisher), NaH$_2$PO$_4$ (Thermo Fisher), and Na$_4$P$_2$O$_7$ (Sigma Aldrich)) with the similar matrix as used for the samples and ranging in concentrations from 0.2 ppb to 100 ppb were analyzed in the same way as the samples. The peak integrals of the standards were used to generate calibration curves, which were then used to quantify concentrations of the four P species in the solution. The detection limits of the IC-ICPMS were < 0.1 ppb for phosphite and phosphate, 0.1 ppb for hypophosphite, and 0.2 ppb for pyrophosphate.

### Phylogenetic tree
Phosphite dehydrogenase genes were obtained from Boden et al.[6]. Briefly, this involved searching for homologs of experimentally-characterised PtxD enzymes in a sample of 865 genomes representing all major orders of the tree of life in GTDB release 95[75]. These sequences were aligned with MAFFT v. 7.4[76], trimmed to remove gaps present in more than 70 % of sequences at a given column with trimAl v1.2rev59[77] and the phylogeny reconstructed in MrBayes v3.2.7a[78] using default parameters plus a mixed amino acid model prior, a proportion of invariant sites and gamma-distributed site rates. Once converged, the resulting tree was rooted with the minimal ancestor deviation method[58]. To differentiate between phosphite dehydrogenases associated with dissimilatory phosphite oxidation which uses phosphite to produce energy and assimilatory phosphite oxidation which uses phosphite as a source of phosphorus (both to support microbial growth), each genome found to harbour a ptxD gene was interrogated for homologs of ptxE, ptdC, ptdG, ptdH, ptdI and ptdF using HMMER3[79] with the scoring thresholds of Ewens et al.[10]. Genomes found to harbour one or more of these homologs are assumed to use their ptxD genes for DPO based on the premise all organisms known to perform DPO harbour one or more of these genes[10,57,80].

## Data availability

All the data generated for this study are provided in the main text or in supplementary materials. The complete data for this study is also available through the National Geoscience Data Centre of the British Geological Survey under https://doi.org/10.5285/dc1d80f5-db1e-42ef-9e07-b3980e43cd43 and https://doi.org/10.5285/2db2945b-1ba8-4324-bf3a-a8f8b0445139.

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

## Acknowledgements

This work was financially supported by a Natural Environment Research Council (NERC < UKRI) Frontiers grant to EES (NE/V010824/1), Marie Skłodowska-Curie Actions grant to ASB (EP/Y026497/1), and Royal Society Wolfson Visiting Fellowship (Royal Society and the Wolfson Foundation) to KK. For the XRD analyses, we acknowledge an Engineering and Physical Science Research Council (EPSRC) Core Equipment Grant (EP/V034138/1) to the School of Chemistry, University of St Andrews. We thank Annabel Long and Oxana Magdysyuk for their help during IC-ICPMS and XRD analyses, respectively. We greatly appreciate the Geological Survey of Western Australia, Perth Core Library, and Rio Tinto core library for providing samples. In order to meet institutional and research funder open access requirements, any accepted manuscript arising shall be open access under a Creative Commons Attribution (CC BY) reuse licence with zero embargo.

## Author contributions

The idea was conceived by E.E.S. and A.S.B.; Rock samples were provided by A.J.B.S., Y.L., and K.K.; Adsorption experiments and IC-ICPMS analysis were performed by A.S.B., and the genomic analysis was done by J.S.B.; The original draft was prepared by A.S.B. with contributions from J.S.B. and A.J.B.S. and was reviewed and edited by all the authors. The project was supervised by E.E.S.

## Competing interests

The authors declare no competing interests.
