## [Transparent Peer Review file · Nature Communications]

Bioavailable phosphite in the surface ocean during the Great Oxygenation Event

Corresponding Author: Dr Abu Baidya

Version 0:

Reviewer comments:

Reviewer #1

(Remarks to the Author)

Questions of how early ocean chemistry fueled and restricted development and activity of early marine microbial organisms are persistent questions. In this study, Baidya and colleagues provide new experimental data and geochemical measurements to constrain the marine concentrations of reduced phosphorus, phosphite, near the time of the Great Oxidation Event, occurring around the time of the Archean-Proterozoic transition. The authors adsorbed phosphite onto hydrous ferric oxide, a presumed precursor mineral to the Banded Iron Formations, which were a significant type of chemical sediment in the oceans at that time. They show that phosphite adsorbs poorly to HFO compared to phosphate, likely due to its lower charge and affinity. They also extracted phosphorus from BIF and determined analytically the fraction of phosphate and phosphite. They use this data to reconstruct possible seawater phosphite concentrations at the GOE/Archean-Proterozoic transition. Citing evidence for the timing of the origin of genes for assimilatory phosphite oxidation near this time, a process that enables microbes to utilize phosphite instead of phosphate, they suggest that this origin was due to the greater availability of phosphite at that time because phosphate would have been removed by increasingly vigorous primary productivity as oxygenic photosynthesis in the surface water.

The experiments and analytics are neatly done. There are some issues with the assumptions of the experiments – that HFO would be the likely primary precipitate. The authors address this, but don't really account for the likely scenario that HFO may simply be a component of the primary minerals to BIF, with other primary minerals also likely. In short, this view of the importance of HFO is too simplistic, and the experiments don't account for the complexity. If HFO formed in shallower waters, more likely those with oxygen, other iron minerals may have formed deeper. As phosphate is a nutrient scavenged type element, it is likely that the phosphate would have been released below the photic zone/thermocline by reductive dissolution and decomposition. Are the estimates from this paper then relevant to the photic zone or the deeper waters? As the proposed mechanism for phosphite production seem to be abiotic/high temperature, these would happen in deeper waters/sediments? Are the BIF sampled for the estimates from shallower or deeper parts of the margin? So what part of the ocean are these estimates relevant for? There isn't really any discussion of biological production of phosphite, but these seems important in modern systems – that it is some kind of product of degradation of phosphonates. But wouldn't the synthesis of these involve microbial phosphate reduction? So it seems to me that the detection of phosphite might reflect a very active microbial redox cycle of phosphorus, rather than some switch over from one source to another. Do the sources/sizes of fluxes to P change at this time? Maybe a bit with enhanced weathering and such, but its not clear if the paper is saying there was less P total due to enhanced sinks (with BIF and PP?) and just more of it was phosphite vs phosphate, or if there were increased sources of phosphite, and if so were they external (more like the abiotic processes mentioned) or internal (like biological cycling?)

So overall I am just feeling a bit confused about the message here. The experiments and measurements are neat and produce an interesting dataset, but I'm just not really convinced that this draws a straight line to the evolution of APO being driven by phosphate limitation in the upper ocean. I think the problem is really with the interpretation. I would suggest to the authors to think a bit more deeply about how the ocean is structured physically, biologically, and chemically, and what part their data represents. Then, what does this say about what was going on at that time? It doesn't always need to be some major biological innovation to be interesting. Maybe it is actually more interesting that such an active biological phosphorus redox cycle was taking place, which would be consistent with a redox, light, and nutrient stratified ocean.

Thank you for the chance to review this manuscript. It caused me to go back and read some older studies more carefully and

I learned a lot about the early and microbial phosphorus cycle.

Figure 4: fix "Geonomic estimate"

Line 243-245: are the requirements mixed up? This sentence seems contradictory.

Line 261-262: I don't see a direct link between the data presented and the statement in this line that P(III) utilization/APO gene evolution was driven by phosphate limitation.

Line 269: if biomass P accumulation was limited, doesn't this contradict the statement in line 260 that increased primary productivity remove P?

Line 280-281: What is the evidence that oceans were depleted in P(V) during/after the GOE, other than "it should have been because of more primary productivity"?

Reviewer #2

(Remarks to the Author)

Review of manuscript NCOMMS-24-80581-T "Geological and experimental evidence of bioavailable phosphite during the Great Oxygenation Event" by Baidya et al.

Overview

In this study the authors investigated the interactions between phosphate, phosphite, and ferric oxyhydroxide minerals to recreate conditions relevant to phosphorus cycling within the Archean ocean. They found a great difference in sorption activity towards the minerals, with phosphite showing very little when compared to phosphate. They compared their experimental findings to phosphite concentrations within the rock record, going into detail describing methods to reconstruct P speciation from dissolving BIF samples. They use this to infer a range of marine phosphite concentrations and proposed the cycling of phosphite mediated by mineral sequestration and microbial uptake.

This paper tells a nice story and addresses a gap in the literature regarding phosphite and its interactions with iron minerals within the Archean Ocean. The authors combine appropriate experimental methods and link their findings to those found within the literature. They conduct an interdisciplinary analysis, linking geochemical data to biochemistry to discuss the far reaching implications of their findings over the development of the early biosphere. An issue with their reconstruction is that they only use one mineral phase HFO (also lacking spectroscopic characterization) to test interactions with phosphite - despite many precursor candidates to BIFs existing, each with different surface chemistry and thus possible reactivity towards the compound of interest. To the authors credit, they do briefly address this point. I would have liked to see a more thorough study of phosphite interactions (e.g. varying pH, incubation time, and mineral phase) however I have not seen much work regarding iron mineral + phosphite chemistry so this paper opens the field up for future work to build upon and address those differences.

As phosphite has recently become a compound of interest to early life chemistry, how it interacts with the environment and various mineral species is of interest to scientific community and readership of Nature Communications.

Specific comments

1. Authors could mention the relative abundances of phosphite vs phosphate in the modern ocean in introduction.
2. Line 103: No characterization method was given for the HFOs used in the experiment. Given that these were precipitated from Fe(II) salts under aerobic conditions in the presence of Si and P compounds, which can affect the rate of Fe(II) oxidation and final mineral phase, how can you be sure what material you are working with especially if you want to replicate them? Personally, I'd be interested in what Fe(II)+phosphite mineral phases look like over time.
3. Figure 1's readability would be improved if each graph was separated and enlarged. Nitpicking, color/shape/size of the points and lines could be improved and there are also inconsistencies with font sizes on the axes.
4. Line 150: It would improve readability if these excluded datapoints from the DI data set omitted and noted in the supplementary as a graph with explanation of why specific points can't be used.
5. Line 168: These other mineral candidates for BIFs all have unique surface sorption properties for phosphate and perhaps too for phosphite. Something for someone to explore later. Linked to this is the role of pH which too could have a strong effect.
6. Line 314: The wording here suggests that NaOH is the oxidizing agent here. NaOH is not an oxidant for Fe²⁺ and all it's doing is precipitating the HFO. I assume you mean that it's the O₂ dissolved in the solution via contact with the atmosphere that is doing the oxidizing- which is unclear in this methods section.
7. Line 315: Is 30 minutes the total incubation time for the experiment? I would find it difficult to conclude the nature of phosphite sorption after only 30 minutes of incubation, especially given crystallization of Fe minerals can be impacted by Si and P content. Did you perform long-term sorption experiments to see if time had an effect?

Reviewer #3

(Remarks to the Author)

Phosphorus has been widely believed to be a bio-limiting nutrient for primary productivity in the long-term during the geological history. Its availability was particularly important for the early Earth studies, due to the efficient uptake of P(IV) by pervasive and enormous precipitation of BIFs. There are many efforts trying to constrain the concentration/fluxes of P(IV) on the early Earth, but very few (however, increasing) studies focus on other phases of phosphorus, including phosphite.

In this study, Baidya combined evidence from geological analyses and experimental simulation and tried to reconstruct availability of phosphite using BIFs. This is a novel work. Moreover, the authors analyzed the phylogenetic data and proposed that increasing availability of phosphite would favor the appearance of P(III)-utilizing metabolisms in the late Archean or paleoproterozoic. This proposal provides more evidence for the study of co-evolution of geosphere and biosphere.

I think this paper is overall of high quality and written well. I have some comments/suggestions for the authors' consideration during their revision.

Line 94 and experiments: the pH of the experiment is not very relevant with early Earth seawater. Previous studies used the pH same to the modern seawater, but recent efforts have shown a lower and circumneutral pH in the late Archean or paleoproterozoic (e.g., Krissansen-Totton et al., 2018, PNAS, 10.1073/pnas.1721296115; Halevy & Bachan, 2017, Science, 10.1126/science.aal4151). I would like to see a sensitivity test by varying the pH.

Line 110-: this is not a salinity effect. This is not the effect of salinity but dissolved silica. There are quite a few studies discussing this effect by environmental scientists, e.g., Chen et al., 2022, EST, 10.1021/acs.est.1c03629.

Figure 3 is confusing. Did you also estimate the concentration of P(IV) in the Archean-Proterozoic boundary?

Line 234: The expression "evolutionary shift" is not explicit. It was more likely that the overall primary producers were still using P(IV) because of its much larger fluxes, but some started using P(III) probably in local P(III)-rich environments.

Line 244: Did you mean "lower", according to the values you showed?

Line 270: I did not get why your results support higher P in the early Archean, which is still controversial. A lower source (BIFs precipitation) did not guarantee higher P(IV). We also need to consider the sources.

One general suggestion:

I suggest to include a discussion on the balance of P(III) sources/sinks and why there was elevation of seawater P(III) in the Archean-Proterozoic boundary. Given the less impact events and hydrothermal activities, the sources might become smaller in the late Archean compared to the early Archean. It is surprising to see a big increase of P(III) in seawater along with time...

A final suggestion: the structure of the manuscript reads a bit confusing. I don't know if it is better but for me, I like to see the geochemical analyses of natural samples firstly and then show the experiments. I will leave this decision to the authors.

Jihua Hao

Version 1:

Reviewer comments:

Reviewer #1

(Remarks to the Author)

The revision of this manuscript largely addresses some minor concerns raised by the reviewers about mineral identity, reaction time, and pH/salinity effects, but I do not think that the major and more conceptual concerns have been adequately addressed. As a result, the messaging of this paper is still very muddled. Furthermore, some revisions of the text have illuminated more inconsistencies and problematic logic in the interpretation of the results. Unfortunately, I think this makes the manuscript unpublishable in its current form. I expand on some justification below.

First, I and another reviewer both raised concerns about the sole focus on HFO (my comment: "The authors address this, but don't really account for the likely scenario that HFO may simply be a component of the primary minerals to BIF, with other primary minerals also likely. In short, this view of the importance of HFO is too simplistic, and the experiments don't account for the complexity.") The authors rebutted that "Our experiments are not relevant to those more reduced BIFs, which we have now emphasized in the text. Our work is relevant to BIFs that formed over the continental shelf settings, i.e. in waters within the photic zone 1,7. We have highlighted this point in the text (lines 196-198)... Therefore, our quantitative estimates are most relevant to shelf and outer shelf settings (and not deepwater settings beyond outer shelf), which we have now emphasized in the revised text (lines 196-198)." While I agree with the authors statement here, the data compilation in Figure 4, particularly the geochemical estimates, are compiled from models and sediments that span surface waters to hydrothermal fluids to deep, and anoxic deep oceans with abundant dissolved Fe²⁺. In fact, the studies cited seem to indicate that surface oceans could have been depleted in P relative to modern while deep oceans could be quite enriched. Unfortunately, these spatial distinctions are not made clear in the manuscript and a temporal distinction is rather emphasized (line 320: "First, although higher concentrations of dissolved P(V) (1-4000 μM) in the early Archean oceans are suggested by genomic⁶ and several geochemical estimates^{25,48,50}, both converge on sub-micromolar concentrations at the onset of the GOE (Figure 4D).") In fact, making this spatially distinction of surface and deep ocean P estimates clearer in the manuscript would actually better support the authors assertion that P could have become a limiting nutrient in surface oceans, particularly during the GOE.

In the section "shift in microbial P-utilization around the GOE", some ranges are given for P(III) in natural systems, in the nM

to low micromolar range. The statement is made that (line 264: “Furthermore, experimental studies suggest that P(III)-dependent microbial growth is possible at 50 μ M P(III)⁷. These concentrations are higher than our estimated concentration of P(III) in seawater at the onset of the GOE, suggesting either local enrichment of P(III) above those calculated averages or lower thresholds of P(III) for the growth of microbial life during the GOE.” I don’t understand how a biological upper limit would be used to make the conclusion that P(III) in seawater at the GOE was elevated above the range indicated by the data. Also, it is not justified why the biological lower limits stated in line 284 would be expected to be different during the GOE. I find it a really logic leap from the data presented for sub-micromolar quantities of P(III) in seawater to the conclusion that P limitation in the upper ocean at the GOE drove the development of P(III)-utilizing organisms. This conclusion should also depend on which organisms utilize P(III), and whether are they are likely to represent the primary producers of the upper ocean (now and in the past) who would have had the highest P demands.

A minor point is that the authors assertion that photoferrotrophs were involved in Fe(II)-oxidation neglects more recent molecular and genetic reconstructions that indicate modern photoferrotrophs might be a very recently evolved lineage. “Shelf-settings were likely oxygenated from the end-Archean onwards⁸ and were the most important habitat of Fe(II)-oxidizing bacteria such as photpferrotrophs. Therefore, the shelf settings were favourable for the precipitation of HFO, facilitated by either free O₂ or photoferrotrophs or both.” See Ward, L. M. & Shih, P. M. Phototrophy and carbon fixation in Chlorobi postdate the rise of oxygen. PLoS One 17, 1–16 (2022). Cardona, T., Sánchez-Baracaldo, P., Rutherford, A. W. & Larkum, A. W. Early Archean origin of Photosystem II. Geobiology 17, 127–150 (2018).

Both I and another reviewer suggest talking about sources/sinks/cycling of P. The authors actually do this in text a bit, and if the text was restructured around the spatial variation of P in the ocean, it would help to identify potential hot spots for production or consumption, even if not quantifying sources and sinks. The authors assert “However, it is beyond the scope of this study to pin down the exact source of phosphite to the ocean around the GOE. Hopefully, our work will stimulate future studies that will quantify different biotic and abiotic source fluxes more accurately.” I think the over-emphasis of a temporal shift to P(III) utilization at the GOE really obscures the discussion of this spatial component that the data presented is really begging be further articulated, and how and where P(III) was produced and where differential microbial P utilization took place.

Reviewer #2

(Remarks to the Author)

The authors neatly addressed the points the reviewers outlined and made changes to the manuscript which improve its argument for the relationships between phosphite, iron minerals, and the greater early biosphere. While constrained to HFOs, I think this paper opens up space for other experiments to explore behaviors between reactive minerals and phosphite chemistry. This paper is worthy of publication.

Reviewer #3

(Remarks to the Author)

The authors have successfully answered all of my concerns. I would like to see the publication of the manuscript as soon as possible.

We thank all the reviewers for reviewing our manuscript and providing constructive feedback. We have provided point-by-point responses to their comments below. In short, we have performed new adsorption experiments for both P(V) and P(III) to constrain the role of pH and experimental duration. Furthermore, we have characterized the solid precipitates using FTIR and XRD. New data support the conclusion that we precipitated hydrous ferric oxides (HFO), most likely as ferrihydrite. Moreover, a change in pH or experimental duration does not quantitatively impact the findings. We have now integrated the new results into the revised manuscript and expanded the discussion to emphasize the above points.

Response to Reviewer #1:

Questions of how early ocean chemistry fueled and restricted development and activity of early marine microbial organisms are persistent questions. In this study, Baidya and colleagues provide new experimental data and geochemical measurements to constrain the marine concentrations of reduced phosphorus, phosphite, near the time of the Great Oxidation Event, occurring around the time of the Archean-Proterozoic transition. The authors adsorbed phosphite onto hydrous ferric oxide, a presumed precursor mineral to the Banded Iron Formations, which were a significant type of chemical sediment in the oceans at that time. They show that phosphite adsorbs poorly to HFO compared to phosphate, likely due to its lower charge and affinity. They also extracted phosphorus from BIF and determined analytically the fraction of phosphate and phosphite. They use this data to reconstruct possible seawater phosphite concentrations at the GOE/Archean-Proterozoic transition. Citing evidence for the timing of the origin of genes for assimilatory phosphite oxidation near this time, a process that enables microbes to utilize phosphite instead of phosphate, they suggest that this origin was due to the greater availability of phosphite at that time because phosphate would have been removed by increasingly vigorous primary productivity as oxygenic photosynthesis in the surface water.

Response: We thank the reviewer for reviewing the manuscript and providing constructive and positive comments. These comments have been helpful to prepare the updated manuscript. We have provided point-by-point response below.

The experiments and analytics are neatly done. There are some issues with the assumptions of the experiments – that HFO would be the likely primary precipitate. The authors address this, but don't really account for the likely scenario that HFO may simply be a component of the primary minerals to BIF, with other primary minerals also likely. In short, this view of the importance of HFO is too simplistic, and the experiments don't account for the complexity.

Response: Experiments are, by necessity, always simplifications of natural systems. Nevertheless, by focusing on HFO precipitation, we believe that we are capturing the dominant phase that is most relevant for the rock samples that we investigated. We have included SEM images of two representative samples to demonstrate that the major iron phases in these are magnetite and hematite (Fig. 1 in this document). HFO is known to be a precursor of these more stable oxides¹. Hence our focus on HFO in the experiments is a valid approach and also in line with previous work^{2,3}.

In terms of the primary precipitates, we are very aware of the push-back from some colleagues that the primary precipitates may have included, or was even dominated, by other iron-bearing phases, such as green rust or greenalite. While we certainly do not discount their possible presence as some fraction of the primary BIF sediment, the argument that some of us keep making (KOK) is that in the presence of an active marine biosphere, it is difficult to imagine

how upwelling Fe^{2+} was not biologically oxidized by either cyanobacterial- O_2 or via photoferrotrophs – certainly by the GOE there is no question that the upper water column was oxygenated. Of course, in deeper waters below the photic zone, greenalite or siderite surely formed, as originally suggested in Konhauser et al.^{1,4-6}. Our experiments are not relevant to those more reduced BIFs, which we have now emphasized in the text. Our work is relevant to BIFs that formed over the continental shelf settings, i.e. in waters within the photic zone^{1,7}. We have highlighted this point in the text (lines 196-198).

If HFO formed in shallower waters, more likely those with oxygen, other iron minerals may have formed deeper. As phosphate is a nutrient scavenged type element, it is likely that the phosphate would have been released below the photic zone/thermocline by reductive

dissolution and decomposition. Are the estimates from this paper then relevant to the photic zone or the deeper waters? As the proposed mechanism for phosphite production seem to be abiotic/high temperature, these would happen in deeper waters/sediments? [this question is addressed below under the next comment.] Are the BIF sampled for the estimates from shallower or deeper parts of the margin? So, what part of the ocean are these estimates relevant for?

Response: We agree with the reviewer that reductive dissolution of iron oxides in deeper waters may release the sorbed P(V) from Fe-phases. However, oxide facies BIFs (i.e., magnetite ± hematite-bearing), including those investigated in this study, have evidently avoided this reductive dissolution process. These types of BIFs are thought to have formed distal to detrital sediment influx from land but near deep-marine upwelling zones that supplied Fe²⁺ (ref.¹). Shelf-settings were likely oxygenated from the end-Archean onwards⁸ and were the most important habitat of Fe(II)-oxidizing bacteria such as photoferrotophs. Therefore, the shelf settings were favourable for the precipitation of HFO, facilitated by either free O₂ or photoferrotophs or both. The preservation of iron oxides in our samples (Fig. 1 in this document) is the evidence of HFO precipitation. So reductive dissolution is not directly relevant to these samples. In the deeper ocean, reductive dissolution could have happened and that could lead to the precipitation of greenalite, green rust, or Fe-carbonate. Therefore, our quantitative estimates are most relevant to shelf and outer shelf settings (and not deepwater settings beyond outer shelf), which we have now emphasized in the revised text (lines 196-198).

There isn't really any discussion of biological production of phosphite, but these seems important in modern systems – that it is some kind of product of degradation of phosphonates. But wouldn't the synthesis of these involve microbial phosphate reduction? So it seems to me that the detection of phosphite might reflect a very active microbial redox cycle of phosphorus, rather than some switch over from one source to another.

Response: Yes, we agree that biological phosphite production could have started by the GOE between 2.5-2.2 Ga. Although no direct biological conversion of P(V) compounds to P(III) is known, biological production of phosphonate from P(V) sources is common in the modern ocean, and P(III) may form due to disintegration of phosphonates, as pointed out by the reviewer. A recent phylogenetic study demonstrated that organisms developed the ability to generate phosphonate during the GOE⁹. Therefore, phosphite production due to phosphonate degradation could have been possible during the GOE. Hence, phosphite may, to some extent, have been produced in the upper ocean by biologic processes – in addition to abiotic processes linked to metamorphism of ferruginous sediments^{10,11} or serpentinization¹². We have emphasized this in the text (lines 254-266). However, it is beyond the scope of this study to pin down the exact source of phosphite to the ocean around the GOE. Hopefully, our work will stimulate future studies that will quantify different biotic and abiotic source fluxes more accurately.

Do the sources/sizes of fluxes to P change at this time? Maybe a bit with enhanced weathering and such, but its not clear if the paper is saying there was less P total due to enhanced sinks (with BIF and PP?) and just more of it was phosphite vs phosphate, or if there were increased sources of phosphite, and if so were they external (more like the abiotic processes mentioned) or internal (like biological cycling?).

Response: The key point of our study is that there was P(III) in the ocean at the onset of the GOE, which has previously been unknown, and it could have been as high as 80% of total dissolved P, implying P(V)-depleting environments, at least in some locales. We have modified the text to make this clear. It is beyond the scope of this study to map out P(III) sources and their evolution through time in detail, but we can speculate that the P(III) was likely derived from a combination of biotic and abiotic sources. We can also speculate that biological phosphite sources may have increased, as noted in response to the previous comment.

It is possible that enhanced oxidative weathering delivered more total phosphorus, but P(III) would likely have undergone oxidation to P(V) during transport. Although oxidative weathering could increase the P(V) supply, a ten-fold increase in biological phosphate consumption¹³ and abiotic phosphate sorption on iron oxides² during BIF precipitation likely removed some, if not the majority of it around the GOE. This could have led to P(V)-depleted environments, at least in some locales. This is in line with our P(III)/P(V) reconstruction during the GOE. In the modern surface ocean, many reduced P species are more commonly used in P(V)-starved settings^{9,14}. Furthermore, we note that life started to utilize reduced P species including P(III) at the onset of the GOE⁹. Hence, we speculate that the driver for increased biological P(III) cycling (both production and utilization) during the GOE may have been phosphate depletion. Hopefully, our findings of the importance of phosphite will stimulate future work on the quantification of source fluxes as well as other possible triggers for the change in P-utilization.

So overall I am just feeling a bit confused about the message here. The experiments and measurements are neat and produce an interesting dataset, but I'm just not really convinced that this draws a straight line to the evolution of APO being driven by phosphate limitation in the upper ocean. I think the problem is really with the interpretation. I would suggest to the authors to think a bit more deeply about how the ocean is structured physically, biologically, and chemically, and what part their data represents. Then, what does this say about what was going on at that time? It doesn't always need to be some major biological innovation to be interesting. Maybe it is actually more interesting that such an active biological phosphorus redox cycle was taking place, which would be consistent with a redox, light, and nutrient stratified ocean.

Response: We have added additional discussion on HFO precipitation in the shallow oxygenated ocean and pointed out that our data may not be used to estimate phosphite concentrations in the deep ocean, where Fe(II)-phases such as greenalite might have precipitated. Regarding the biological P(III) supply, we have pointed out in the revised text that this is a valid mechanism for generating additional P(III) and a combination of abiotic and biotic processes could have delivered P(III) at the time of the GOE.

We stand by our point that it is likely that there was some pressure on the biosphere due to depletion of P(V) by a combination of biological demand and adsorption onto HFO at the onset of GOE (>2.5 Ga). We further believe that this may have led to new metabolic strategies of P utilization, including P(III) and phosphonate utilization. In the modern surface ocean, many reduced phosphorus species are more widely used in areas where phosphate is limiting^{9,14}. Similar P-limiting environments could have existed at the onset of the GOE, which is evident by several geochemical and genomic data. First, some recent geochemical and genomic estimates suggest that P(V) concentrations in the Archean ocean could have reached to tens to thousands of μM ^{4,15,16}, however, that concentration came down to ten to hundreds of times less than one μM during the GOE as evident in both geochemical and genomic estimates⁹. Second, our geochemical data suggest that at the onset of the GOE, at places in the ocean,

as much as 80% of the total dissolved phosphorus might have been P(III), implying a P(V) scarcity. Third, our genomic data suggest that assimilatory phosphite oxidation or APO evolved at this time, a mechanism in which microorganism convert P(III) into P(V) before utilizing it. Such P utilization mechanisms are not likely to be required if P(V) is abundant in the first place. For example, in the modern surface oceans, comparatively more organisms have phosphonate assimilation genes in environments having a scarcity of P(V)^{9,14}. Hence, even if our data may reflect microbial P redox cycling, they are still most consistent with phosphate limitation.

Thank you for the chance to review this manuscript. It caused me to go back and read some older studies more carefully and I learned a lot about the early and microbial phosphorus cycle.

Figure 4: fix "Geonomic estimate".

Response: We have fixed the spelling in Figure 4 in the main manuscript file.

Line 243-245: are the requirements mixed up? This sentence seems contradictory.

Response: Yes, the concentrations got mixed up. We have corrected them.

Line 261-262: I don't see a direct link between the data presented and the statement in this line that P(III) utilization/APO gene evolution was driven by phosphate limitation.

Response: We have added additional discussion to emphasize this point. Please see above.

Line 269: if biomass P accumulation was limited, doesn't this contradict the statement in line 260 that increased primary productivity remove P?

Response: We have deleted the phrase as it was creating confusion. Our main point here is that at least a part of dissolved P(V) would have been removed by ferrihydrite during BIF precipitation. The revised text highlights that.

Line 280-281: What is the evidence that oceans were depleted in P(V) during/after the GOE, other than "it should have been because of more primary productivity"?

Response: Our P(III) and genomic data collectively indicate that at least some locales in the ocean were depleted in P(V) at the time of GOE. Our geochemical data suggest that, on average, more than 45% (reaching up to >80%) of total dissolved P was P(III) during deposition of the Dales Gorge, Marra Mamba, and the Kuruman iron formations at the onset of GOE. Our genomic data point to the evolution of APO at that time, a mechanism of P-utilization that does not happen in the presence of abundant P(V) in modern ocean¹⁴. We therefore suggest that there was a P(V) limitation in these basins.

Response to Reviewer #2:

Overview

In this study the authors investigated the interactions between phosphate, phosphite, and ferric oxyhydroxide minerals to recreate conditions relevant to phosphorus cycling within the Archean ocean. They found a great difference in sorption activity towards the minerals, with phosphite showing very little when compared to phosphate. They compared their experimental findings to phosphite concentrations within the rock record, going into detail describing methods to reconstruct P speciation from dissolving BIF samples. They use this to infer a range of marine phosphite concentrations and proposed the cycling of phosphite mediated by mineral sequestration and microbial uptake.

This paper tells a nice story and addresses a gap in the literature regarding phosphite and its interactions with iron minerals within the Archean Ocean. The authors combine appropriate experimental methods and link their findings to those found within the literature. They conduct an interdisciplinary analysis, linking geochemical data to biochemistry to discuss the far reaching implications of their findings over the development of the early biosphere. An issue with their reconstruction is that they only use one mineral phase HFO (also lacking spectroscopic characterization) to test interactions with phosphite - despite many precursor candidates to BIFs existing, each with different surface chemistry and thus possible reactivity towards the compound of interest. To the authors credit, they do briefly address this point. I would have liked to see a more thorough study of phosphite interactions (e.g. varying pH, incubation time, and mineral phase) however I have not seen much work regarding iron mineral + phosphite chemistry so this paper opens the field up for future work to build upon and address those differences.

As phosphite has recently become a compound of interest to early life chemistry, how it interacts with the environment and various mineral species is of interest to scientific community and readership of Nature Communications.

Response: We thank the reviewer for providing constructive and positive comments on our manuscript. These comments have been helpful to prepare the updated version. We have provided response to specific comments below.

We have added XRD and FTIR analyses of the precipitate and undertaken additional experiments at $\text{pH } 6.75 \pm 0.25$ and for extended periods of time (24 hours in addition to 30 minutes), also in line with Reviewer 3.

The XRD and FTIR data collectively suggest that the Fe-precipitate is HFO in our experiments. The FTIR data of the precipitate (Fe in seawater without Si) show absorption peaks at 3360-3370 and 1641-1645 cm^{-1} , which corresponds to OH-stretching and OH-bending, respectively, suggesting that the precipitate has OH in its structure (Fig. 2A, B in this document)^{17,18}. The XRD data show broad peaks at ~ 32.4 and 60.5 , which are characteristics of ferrihydrite (Fig. 2C in this document)^{19,20}. We therefore confirm that the Fe-phase that has precipitated in our experiments is hydrous ferric oxide (HFO), most likely ferrihydrite. In experiments containing Si, a mixture and amorphous SiO_2 and HFO has precipitated (Fig. 2C in this document). The precipitation of amorphous silica is likely as we have used 2.2 mM Si in the experimental solution, which is the saturation limit of amorphous silica in seawater³. Amorphous silica is identified in XRD and FTIR data. Previous studies noted that the presence of Si in Fe-Si- H_2O systems like ours may shift the ferrihydrite XRD peaks towards amorphous silica^{19,20}. We see the similar effect in P(V) adsorption experimental products (Fig. 2D in this document). Despite this shift, the major ferrihydrite peak at ~ 32.5 is observed in P(III) experimental products (Fig. 2C in this document). In summary, HFO precipitates in seawater without Si and a mixture of

amorphous silica and HFO precipitates in seawater containing Si at our experimental conditions.

We see minor or no effects of pH and experimental duration on P(V) and P(III) adsorption at our experimental conditions. We conducted new adsorption experiments at $\text{pH } 6.75 \pm 0.25$ and tested two different duration times, 0.5 and 24 hours (Fig. 3C in this document). In case of 24-hour experiments, the pH was monitored after 0.5, 1, and 20 hours before collecting the samples. The major trend of P(V) and P(III) adsorption at $\text{pH } 6.75 \pm 0.25$ is similar to that at $\text{pH } 8 \pm 0.2$, i.e., the sorption of P(III) is limited compared to P(V) (Fig. 3C in this document). We note that K_{ads} value for P(V) at $\text{pH } 6.75 \pm 0.25$ is 0.026, which is very similar to what Jones et al.² reported for P(V) adsorption at pH 8 ($K_{\text{ads}} = 0.021$) (Fig. 3C, D in this document). Similarly for P(III), the K_{ads} values at $\text{pH } 6.75 \pm 0.25$ and 8 ± 0.2 are 0.0005 and 0.0008, respectively, which is within the error limit of the instrument (Fig. 3B, C in this document). This suggests that the control of pH on P(III) and P(V) adsorption at pH 6.75-8 is negligible. Furthermore, we note that for 0.5-hour and 24-hour P(V) adsorption experiments, the K_{ads} changed from 0.026 to 0.027, implying a negligible effect of experimental duration. K_{ads} values for P(III) for the same duration times are indistinguishable (Fig. 3C in this document). This implies minor or no effect of duration on P(V) and P(III) adsorption at the experimental conditions. This is consistent with Jones et al.², who also found that the effect of duration for P(V) adsorption at pH 8 is limited. We therefore conclude that our results remain valid, even if the pH of the ancient ocean was lower than it is today, and that adsorption had reached equilibrium after 0.5 hours.

Figure 2: Representative FTIR (A, B) and XRD (C, D) data of precipitates from different experiments. In 'Si in Seawater' and 'Fe(II) in Seawater' experiments, Si and Fe (II) were added to the seawater solutions. The precipitates had white and red/reddish brown color, respectively. Adding only Si and only Fe(II) in seawater precipitate silica and ferrihydrite, respectively (C, D). Broad peaks of precipitated solids in these two experiments suggest that they are amorphous (C, D). Both these phases have OH in their structures as seen in A and B, suggesting that hydroxyferrihydrite (HFO) is the Fe-phase. 'Sea-Si' stands for seawater with 2.2 mM Si. FTIR pattern of the adsorption experimental samples is dominated by amorphous silica. In C and D, halite (NaCl, blue squares) is identified, however, it has precipitated during freeze-drying of the experimental precipitates. The concentration of NaCl (0.56M) is not enough to precipitate halite during the experiment. In P(III) adsorption experiments, both amorphous silica and HFO are identified (C).

We have expanded the discussion in the revised manuscript based on these new results.

As noted in response to a similar comment from Reviewer 1, experiments are by necessity simplifications of natural systems, but we believe that our focus on HFO is valid for the BIFs analysed in this study, as those are largely dominated by iron oxide phases (Fig. 1 in this document). Our results are applicable to outer-shelf BIFs, where either free O₂ or photoferrotrophs or both could have facilitated Fe(II) oxidation and HFO precipitation. Therefore, the P(III) estimations are valid for shelf and outer-shelf settings at the onset of the GOE. We have now emphasized this in the revised manuscript.

We thank the reviewer for acknowledging the pioneering character of our study that will hopefully stimulate future work in the direction of phosphite sinks (and sources) from the ocean.

Specific comments

1. Authors could mention the relative abundances of phosphite vs phosphate in the modern ocean in introduction.

Response: Phosphite concentration is not known in the modern ocean²¹. One study by Mooy et al.²¹ attempted to measure P(III) concentration in tropical North Atlantic ocean but did not detect any (where their detection limit was 1.5 μM).

2. Line 103: No characterization method was given for the HFOs used in the experiment. Given that these were precipitated from Fe(II) salts under aerobic conditions in the presence of Si and P compounds, which can affect the rate of Fe(II) oxidation and final mineral phase, how can you be sure what material you are working with especially if you want to replicate them?

Response: Our new XRD and FTIR data collectively indicate that the precipitated Fe-phase is indeed HFO.

Personally, I'd be interested in what Fe(II)+phosphite mineral phases look like over time.

Response: Only 2-5% of initial P(III) precipitated along with Fe(II), meaning ferrous phosphite is not a major sink of phosphite in our experiments (Fig. 3 in this document). XRD and FTIR did not detect any ferrous phosphite phases. Even if some new ferrous phosphite phases precipitated, those would be very difficult to detect using XRD or other spectrometric methods because the quantity is very small.

3. Figure 1's readability would be improved if each graph was separated and enlarged. Nitpicking, color/shape/size of the points and lines could be improved and there are also inconsistencies with font sizes on the axes.

Response: We have added the newly generated data on P(V) and P(III) adsorption in a separate graph and added to Figure 1 in the main manuscript file. We have now used the same font for all the text in the graphs. We have changed the colour of the legends for better readability.

4. Line 150: It would improve readability if these excluded datapoints from the DI data set omitted and noted in the supplementary as a graph with explanation of why specific points can't be used.

Response: We have now added all the datapoints in a Supplementary Figure and mentioned which datapoint we have not included due to experimental error. Only the useful datapoints are kept in the graph presented in the main text.

5. Line 168: These other mineral candidates for BIFs all have unique surface sorption properties for phosphate and perhaps too for phosphite. Something for someone to explore later.

Response: Yes, we agree that this should be explored in the future. HFO is a valid focus for oxidized BIFs, which is the case for our samples (Fig. 1 in this document).

Linked to this is the role of pH which too could have a strong effect.

Response: New experiments suggest that pH has limited control on the P(III) and P(V) sorption onto HFO within the tested range of pH 6.5-8. See Fig. 3C above.

6. Line 314: The wording here suggests that NaOH is the oxidizing agent here. NaOH is not an oxidant for Fe²⁺ and all it's doing is precipitating the HFO. I assume you mean that it's the O₂ dissolved in the solution via contact with the atmosphere that is doing the oxidizing- which is unclear in this methods section.

Response: Yes, the reviewer is right that NaOH is not the oxidant. In the alkaline conditions, atmospheric O₂ is the oxidant. We have made this clear in the method section.

7. Line 315: Is 30 minutes the total incubation time for the experiment? I would find it difficult to conclude the nature of phosphite sorption after only 30 minutes of incubation, especially given crystallization of Fe minerals can be impacted by Si and P content. Did you perform long-term sorption experiments to see if time had an effect?

Response: We acknowledge the concern of the reviewer here, and to address it we performed new experiments. We also see that the term '30-minute duration' was a bit confusing for the readers. In reality, after adding the Fe(II) to the experimental solution, we added NaOH to create the desired pH. It usually took 2-5 minutes to stabilize the pH and then we maintained the pH for 30 minutes by adding diluted NaOH/HCl. In the revised manuscript, we have made this clear in the method section. In the new experiments, we have considered 30 minutes as before, and we have added new 24-hour experiments to see how the experimental duration affects the adsorption. It can be seen in Fig. 3C above that duration did not have any detectable effect on P(III) adsorption, indicating that equilibrium was already reached after 30 minutes. Duration did not have a significant effect for P(V) adsorption either (Fig. 3C in this document). A negligible effect of duration on P(V) sorption is also reported in a previous study². We therefore suggest that the duration has no significant impact on P(III) and P(V) adsorption, particularly in the considered experimental conditions. Our experiments appear to capture equilibrium conditions.

Response to Reviewer #3:

Phosphorus has been widely believed to be a bio-limiting nutrient for primary productivity in the long-term during the geological history. Its availability was particularly important for the early Earth studies, due to the efficient uptake of P(IV) by pervasive and enormous precipitation of BIFs. There are many efforts trying to constrain the concentration/fluxes of P(IV) on the early Earth, but very few (however, increasing) studies focus on other phases of phosphorus, including phosphite.

In this study, Baidya combined evidence from geological analyses and experimental simulation and tried to reconstruct availability of phosphite using BIFs. This is a novel work. Moreover, the authors analyzed the phylogenetic data and proposed that increasing availability of phosphite would favor the appearance of P(III)-utilizing metabolisms in the late Archean or paleoproterozoic. This proposal provides more evidence for the study of co-evolution of geosphere and biosphere.

I think this paper is overall of high quality and written well. I have some comments/suggestions for the authors' consideration during their revision.

Response: We thank the reviewer for providing constructive and positive comments on our manuscript. These comments and suggestions have been helpful to prepare the updated version. We have provided response to specific comments below.

Line 94 and experiments: the pH of the experiment is not very relevant with early Earth seawater. Previous studies used the pH same to the modern seawater, but recent efforts have shown a lower and circumneutral pH in the late Archean or paleoproterozoic (e.g., Krissansen-Totton et al., 2018, PNAS, 10.1073/pnas.1721296115; Halevy & Bachan, 2017, Science, 10.1126/science.aal4151). I would like to see a sensitivity test by varying the pH.

Response: As suggested by the reviewer, we did a pH sensitivity test at pH 6.75 ± 0.25 . The new data are shown in Fig. 3C in this document. We note that K_{ads} value for P(V) at pH 6.75 ± 0.25 is 0.026, which is very similar to Jones et al.² who reported a K_{ads} value of 0.021 for P(V) adsorption at pH 8. Similarly, the K_{ads} values for P(III) adsorption at pH 6.75 ± 0.25 and 8 ± 0.2 are 0.0005 and 0.0008, respectively, which are within the error limit of the method. This suggests that the control of pH on P(III) and P(V) adsorption at pH 6.75-8 is negligible. Importantly, the large difference between P(V) and P(III) adsorption at pH 6.75 ± 0.25 is similar to that at pH 8 ± 0.2 , i.e., the sorption of P(III) onto HFO is much less compared to P(V). Therefore, our main conclusions remain the same. We have expanded the discussion in the revised manuscript based on the new results.

Line 110-: this is not a salinity effect. This is not the effect of salinity but dissolved silica. There are quite a few studies discussing this effect by environmental scientists, e.g., Chen et al., 2022, EST, 10.1021/acs.est.1c03629.

Response: We agree with the reviewer that dissolved silica affects P(V) adsorption onto HFO such that the presence of dissolved silica reduces the P(V) adsorption (e.g., ref 3). In that case, dissolved silica would give lower K_{ads} values compared to silica-free solution, if other conditions remain the same. In our case, we see that K_{ads} values in DI water, 10-times diluted seawater with 0.22 mM Si, and seawater with 2.2 mM Si are 0.011, 0.039, and 0.021, respectively. It therefore seems that despite the presence of dissolved silica, K_{ads} values are higher in diluted seawater and seawater compared to DI water. Hence, we think that salinity

has some control on P(V) adsorption onto HFO. We have rewritten this section in the manuscript to emphasize this point.

Figure 3 is confusing. Did you also estimate the concentration of P(IV) in the Archean-Proterozoic boundary?

Response: We are not sure what is meant by P(IV), we think that the reviewer meant P(V). If so, yes, we did estimate the P(V) concentrations as well. We have explicitly mentioned this in the revised manuscript.

Line 234: The expression "evolutionary shift" is not explicit. It was more likely that the overall primary producers were still using P(IV) because of its much larger fluxes, but some started using P(III) probably in local P(III)-rich environments.

Response: Yes, we agree with the reviewer that most primary producers were still using P(V) and those in the P(V)-depleted environments likely started utilizing alternative P species such as P(III). We have changed the wording to emphasize this.

Line 244: Did you mean "lower", according to the values you showed?

Response: Yes, correct. This has been amended in the MS.

Line 270: I did not get why your results support higher P in the early Archean, which is still controversial. A lower source (BIFs precipitation) did not guarantee higher P(IV). We also need to consider the sources.

Response: Our language here was not clear, which led to this confusion. We estimated P(III) and P(V) in the ocean using the BIF P(III) and P(V) data, and the result is indicative of sub-micromolar concentrations of both P(III) and P(V) at the onset of the GOE (Figure 5 in the main text). The higher concentrations of P(V) in the Archean ocean were estimated by previous geochemical methods and a recent genomic method. We have changed the wording to make this clear.

One general suggestion: I suggest to include a discussion on the balance of P(III) sources/sinks and why there was elevation of seawater P(III) in the Archean-Proterozoic boundary. Given the less impact events and hydrothermal activities, the sources might become smaller in the late Archean compared to the early Archean. It is surprising to see a big increase of P(III) in seawater along with time.

Response: The relative proportions of P(III) sources and sinks and their balance in the Precambrian are largely unknown. Major source of P(III) in the Archean may include meteoritic delivery²², lightning-induced formation of phosphides^{23,24}, metamorphism of ferruginous sediments^{10,11}, and serpentinization¹². During the GOE, meteoritic delivery might have been limited but the other abiotic sources could have been active. However, the relative proportions of these phases are unknown. Furthermore, there could be some biologic P(III) in the ocean at this time. In the modern ocean, some microorganisms convert P(V) into phosphonate, which later decomposes into P(III). A genomic study⁹ suggests that phosphonate producers

appeared during the GOE, therefore, some P(III) at this time could have been partially biogenic as a result of the disintegration of biologically-produced phosphonates. We have modified the discussion to capture this complexity.

A final suggestion: the structure of the manuscript reads a bit confusing. I don't know if it is better but for me, I like to see the geochemical analyses of natural samples firstly and then show the experiments. I will leave this decision to the authors.

Jihua Hao

Response: We prefer to keep the order as it is. As there was no study on P(III) sorption onto HFO, we first quantify the difference between P(III) and P(V) sorption. We note that P(III) sorption is fundamentally different from P(V) sorption. This is valid even if we did not see any P(III) in the studied BIF samples.

References

1. Konhauser, K. O. *et al.* Iron formations: A global record of Neoproterozoic to Palaeoproterozoic environmental history. *Earth-Science Rev.* **172**, 140–177 (2017).
2. Jones, C., Nomosatryo, S., Crowe, S. A., Bjerrum, C. J. & Canfield, D. E. Iron oxides, divalent cations, silica, and the early earth phosphorus crisis. *Geology* **43**, 135–138 (2015).
3. Konhauser, K. O., Lalonde, S. V., Amskold, L. & Holland, H. D. Was there really an Archean phosphate crisis? *Science (80-.)*. **315**, 1234 (2007).
4. Rasmussen, B., Muhling, J. R. & Tosca, N. J. Nanoparticulate apatite and greenalite in oldest, well-preserved hydrothermal vent precipitates. *Sci. Adv.* **10**, eadj4789 (2024).
5. Rasmussen, B., Muhling, J. R. & Krapež, B. Greenalite and its role in the genesis of early Precambrian iron formations – A review. *Earth-Science Rev.* **217**, 103613 (2021).
6. Konhauser, K. O. *et al.* Decoupling photochemical Fe(II) oxidation from shallow-water BIF deposition. *Earth Planet. Sci. Lett.* **258**, 87–100 (2007).
7. Konhauser, K. O., Kappler, A., Lalonde, S. V. & Robbins, L. J. Trace elements in iron formation as a window into biogeochemical evolution accompanying the oxygenation of Earth's atmosphere. *Geosci. Canada* **50**, 239–258 (2023).
8. Ostrander, C. M. *et al.* Fully oxygenated water columns over continental shelves before the Great Oxidation Event. *Nat. Geosci.* **12**, 186–191 (2019).
9. Boden, J. S., Zhong, J., Anderson, R. E. & Stüeken, E. E. Timing the evolution of phosphorus-cycling enzymes through geological time using phylogenomics. *Nat. Commun.* **15**, 3703 (2024).
10. Herschy, B. *et al.* Archean phosphorus liberation induced by iron redox geochemistry. *Nat. Commun.* **9**, 1346 (2018).
11. Baidya, A. S., Pasek, M. A. & Stüeken, E. E. Moderate and high-temperature metamorphic conditions produced diverse phosphorous species for the origin of life. *Commun. Earth Environ.* **Accepted**, (2024).
12. Pasek, M. A. *et al.* Serpentinization as a route to liberating phosphorus on habitable worlds. *Geochim. Cosmochim. Acta* **336**, 332–340 (2022).
13. Crockford, P. W., Bar On, Y. M., Ward, L. M., Milo, R. & Halevy, I. The geologic history of primary productivity. *Curr. Biol.* **33**, 4741–4750.e5 (2023).
14. Lockwood, S., Greening, C., Baltar, F. & Morales, S. E. Global and seasonal variation of marine phosphonate metabolism. *ISME J.* **16**, 2198–2212 (2022).
15. Crockford, P. & Halevy, I. Questioning the paradigm of a phosphate-limited Archean biosphere. *Geophys. Res. Lett.* **49**, e2022GL099818 (2022).
16. Brady, M. P., Tostevin, R. & Tosca, N. J. Marine phosphate availability and the chemical

- origins of life on Earth. *Nat. Commun.* **13**, 5162 (2022).
17. Vaughan, G., Brydson, R. & Brown, A. Characterisation of Synthetic Two-line Ferrihydrite by Electron Energy Loss Spectroscopy. *J. Phys. Conf. Ser.* **371**, 12079 (2012).
 18. Wang, H. *et al.* Adsorption capacities of poorly crystalline Fe minerals for antimonate and arsenate removal from water: adsorption properties and effects of environmental and chemical conditions. *Clean Technol. Environ. Policy* **20**, 2169–2179 (2018).
 19. Xu, Z., Yu, J. & Xiao, W. Microemulsion-Assisted Preparation of a Mesoporous Ferrihydrite/SiO₂ Composite for the Efficient Removal of Formaldehyde from Air. *Chem. – A Eur. J.* **19**, 9592–9598 (2013).
 20. Xiang, X. *et al.* Hydrolysis Products of Fe(III)-Si Systems With Different Si/(Si + Fe) Molar Ratios: Implications to Detection of Ferrihydrite on Mars. *J. Geophys. Res. Planets* **129**, e2023JE008031 (2024).
 21. Van Mooy, B. A. S. *et al.* Major role of planktonic phosphate reduction in the marine phosphorus redox cycle. *Science (80-.)*. **348**, 783–785 (2015).
 22. Pasek, M. A. & Lauretta, D. S. Aqueous corrosion of phosphide minerals from iron meteorites: A highly reactive source of prebiotic phosphorus on the surface of the early Earth. *Astrobiology* **5**, 515–535 (2005).
 23. Pasek, M. & Block, K. Lightning-induced reduction of phosphorus oxidation state. *Nat. Geosci.* **2**, 553–556 (2009).
 24. Hess, B. L., Piazzolo, S. & Harvey, J. Lightning strikes as a major facilitator of prebiotic phosphorus reduction on early Earth. *Nat. Commun.* **12**, 1535 (2021).

We thank the reviewers for their comments and suggestions on our revised manuscript. We are glad to see that Reviewers #2 and #3 are satisfied with the revision and did not have additional comments. We have provided point-by-point responses to comments from Reviewer #1 below.

Response to Reviewer #1:

The revision of this manuscript largely addresses some minor concerns raised by the reviewers about mineral identity, reaction time, and pH/salinity effects, but I do not think that the major and more conceptual concerns have been adequately addressed. As a result, the messaging of this paper is still very muddled. Furthermore, some revisions of the text have illuminated more inconsistencies and problematic logic in the interpretation of the results. Unfortunately, I think this makes the manuscript unpublishable in its current form. I expand on some justification below.

(1) First, I and another reviewer both raised concerns about the sole focus on HFO (my comment: “The authors address this, but don’t really account for the likely scenario that HFO may simply be a component of the primary minerals to BIF, with other primary minerals also likely. In short, this view of the importance of HFO is too simplistic, and the experiments don’t account for the complexity.”) The authors rebutted that “Our experiments are not relevant to those more reduced BIFs, which we have now emphasized in the text. Our work is relevant to BIFs that formed over the continental shelf settings, i.e. in waters within the photic zone 1,7. We have highlighted this point in the text (lines 196-198)... Therefore, our quantitative estimates are most relevant to shelf and outer shelf settings (and not deepwater settings beyond outer shelf), which we have now emphasized in the revised text (lines 196-198).” While I agree with the authors statement here, the data compilation in Figure 4, particularly the geochemical estimates, are compiled from models and sediments that span surface waters to hydrothermal fluids to deep, and anoxic deep oceans with abundant dissolved Fe²⁺. In fact, the studies cited seem to indicate that surface oceans could have been depleted in P relative to modern while deep oceans could be quite enriched. Unfortunately, these spatial distinctions are not made clear in the manuscript and a temporal distinction is rather emphasized (line 320: “First, although higher concentrations of dissolved P(V) (1-4000 μM) in the early Archean oceans are suggested by genomic⁶ and several geochemical estimates^{25,48,50}, both converge on sub-micromolar concentrations at the onset of the GOE (Figure 4D).”) In fact, making this spatially distinction of surface and deep ocean P estimates clearer in the manuscript would actually better support the authors assertion that P could have become a limiting nutrient in surface oceans, particularly during the GOE.

Response: The reviewer is correct that the compiled data in Fig. 4 consists of phosphate estimation in a wide range of ocean environments including shallow sea, deep ocean, and hydrothermal vents. We note that Brady et al. (2022)¹ estimated the phosphate concentration to be 200-4000 μM in the Hadean/early Archean ocean. However, they did not provide spatial constraints meaning this concentration could be in either or both shallow and deep water. Their inferences are based on laboratory experiments and not tied to a particular sedimentary facies from the rock record. In contrast, several other studies based on deep-water rocks such as jaspilite and vent precipitates specifically pointed out 5-65 times higher P(V) concentrations in deep

Archean water compared to modern deepwater^{2,3}. Furthermore, using carbonate chemistry, Ingalls et al. (2022) estimated P(V) concentration in Neoproterozoic surface and deep water to be 4-12 times and 5-50 times greater, respectively, than modern deep ocean water (2.3 μM). It therefore seems that the surface ocean would have been depleted in P(V) compared to the deep ocean, particularly in the late Archean, in agreement with the reviewer's comment and the general notion that primary productivity in the photic zone lowers the dissolved P concentration in marginal marine settings. We have modified the discussion accordingly emphasizing on spatial distribution of phosphite in ocean at the onset of the GOE (lines 209-213 and 335-342). We now make a clear distinction between deep anoxic water and shallow water to emphasize that our phosphite estimation is more relevant for shallow water (outer shelf).

(2) In the section "shift in microbial P-utilization around the GOE", some ranges are given for P(III) in natural systems, in the nM to low micromolar range. The statement is made that (line 264: "Furthermore, experimental studies suggest that P(III)-dependent microbial growth is possible at 50 μM P(III)⁷. These concentrations are higher than our estimated concentration of P(III) in seawater at the onset of the GOE, suggesting either local enrichment of P(III) above those calculated averages or lower thresholds of P(III) for the growth of microbial life during the GOE." I don't understand how a biological upper limit would be used to make the conclusion that P(III) in seawater at the GOE was elevated above the range indicated by the data. Also, it is not justified why the biological lower limits stated in line 284 would be expected to be different during the GOE. I find it a really logic leap from the data presented for sub-micromolar quantities of P(III) in seawater to the conclusion that P limitation in the upper ocean at the GOE drove the development of P(III)-utilizing organisms. This conclusion should also depend on which organisms utilize P(III), and whether are they are likely to represent the primary producers of the upper ocean (now and in the past) who would have had the highest P demands.

Response: The reviewer is correct that we do not have sufficient evidence in favour of higher than estimated concentrations of phosphite at the onset of GOE. Our data also do not confirm if microbial life had lower phosphite thresholds at that time. Hence, we have deleted sentences that discussed these possibilities. We note that our estimated concentrations (0.00-0.17 μM) coincide with the lower end of phosphite ranges in natural environments including open ocean⁴, geothermal pools (0.06 \pm 0.02 μM)⁵, lakes (0.01-0.71 μM)^{6,7}, rivers (0.08-0.9 μM)⁸, and ponds (0.14-2.90 μM)⁸, some of which (e.g., surface ocean, deep ocean and hydrothermal vents) are inhabited by microbial life that utilize phosphite^{9,10}. Therefore, we believe that our estimated concentrations of phosphite in shallow seawater were sufficient to support microbial growth, particularly when phosphate was limited. We have edited the section accordingly (lines 265-269).

(3) A minor point is that the authors assertion that photoferrotrophs were involved in Fe(II)-oxidation neglects more recent molecular and genetic reconstructions that indicate modern photoferrotrophs might be a very recently evolved lineage. "Shelf-settings were likely oxygenated from the end-Archean onwards⁸ and were the most important habitat of Fe(II)-oxidizing bacteria such as photoferrotrophs. Therefore, the

shelf settings were favourable for the precipitation of HFO, facilitated by either free O₂ or photoferrotrophs or both.” See Ward, L. M. & Shih, P. M. Phototrophy and carbon fixation in Chlorobi postdate the rise of oxygen. *PLoS One* 17, 1–16 (2022). Cardona, T., Sánchez-Baracaldo, P., Rutherford, A. W. & Larkum, A. W. Early Archean origin of Photosystem II. *Geobiology* 17, 127–150 (2018).

Response: We appreciate the reviewer’s comment. However, we respectfully disagree with the assumption that the absence of a molecular clock signal equates to the absence of a metabolism. Molecular clocks cannot account for extinct lineages or undiscovered extant species with deeper evolutionary roots. While molecular data suggest that photoferrotrophs emerged after the GOE, this does not preclude the existence of similar metabolisms earlier, particularly if anoxygenic photosynthesis predates oxygenic photosynthesis. Similarly, recent work by Sanchez-Baracaldo¹¹ suggests that planktonic cyanobacteria evolved only in the Neoproterozoic. If we accept both the Ward and Sanchez-Baracaldo studies, this would imply an absence of planktonic cyanobacteria and photoferrotrophs before the GOE. This presents a challenge, given the widely accepted role of biological activity in BIF deposition and the necessity of planktonic organisms for generating high organic carbon shales in the Archean. By the same logic, interpreting the geologic record this way would suggest that the absence of sandstones at 3.9 Ga implies no continental weathering, no rivers, and even no water—an interpretation at odds with other lines of evidence.

(4) Both I and another reviewer suggest talking about sources/sinks/cycling of P. The authors actually do this in text a bit, and if the text was restructured around the spatial variation of P in the ocean, it would help to identify potential hot spots for production or consumption, even if not quantifying sources and sinks. The authors assert “However, it is beyond the scope of this study to pin down the exact source of phosphite to the ocean around the GOE. Hopefully, our work will stimulate future studies that will quantify different biotic and abiotic source fluxes more accurately.” I think the over-emphasis of a temporal shift to P(III) utilization at the GOE really obscures the discussion of this spatial component that the data presented is really begging to be further articulated, and how and where P(III) was produced and where differential microbial P utilization took place.

Response: We agree with the reviewer that the questions regarding sources/sinks/cycling of phosphite are crucial for better understanding of P cycle in the Archean and Proterozoic. However, we have to acknowledge that numerous constraints are still lacking, even for the modern ocean. We have added a discussion on phosphite sources and sinks based on available information, but this is limited by necessity. Please see lines 271-288. We have mentioned clearly both the known organic and inorganic sources of phosphite and their potential relevance at the time of the GOE, but it is not possible to provide accurate quantitative constraints. As far as sinks of phosphite are concerned, these are also largely uncharacterized. We can be confident that some phosphite was used biologically, as suggested by genomic data¹², and that some phosphite was incorporated into iron minerals, as shown by our own data. However, numerous other potential sinks have not been measured, prohibiting an accurate reconstruction of total sink fluxes. Although our data do not permit further

elaborate discussion on the sources and sinks of phosphite during the GOE, we think that our work is nevertheless a valuable contribution that emphasizes the importance of phosphite on the early Earth and will hopefully stimulate further research on this topic, as also suggested by Reviewer 2.

Our data also allow us to place the first ever quantitative constraints on the concentration of phosphite in the shallow ocean (up to the outer shelf) as the BIF that we studied precipitated in this environment. This may further indicate that the microbial life that lived in the photic zone were able to utilize phosphite. These two points are now made clear in the 'Shift in microbial P-utilization around the GOE' section. However, which sources (such as serpentinization, metamorphism of ferruginous sediments, meteoritic supply, and biological sources) contributed to this phosphite remain unknown and will require additional study. Phosphite is kinetically stable in water and may have a half-life of more than half a million years in the absence of biological uptake^{13,14}. The half-life in the presence of biology is unknown. But a long half-life means that phosphite could be well mixed throughout the ocean. Several sources may have contributed to phosphite both in shallow water, as we note in this study, and deep water, if there was phosphite. The science behind phosphite is relatively new (this is also pointed out by Reviewer 2) and we have very limited data on its sources/sinks in the Precambrian. Even for the much more widely studied phosphate there is still much to learn about sources, sinks and reservoir sizes. Hence, although questions such as (i) how and where P(III) was produced and (ii) where did microbial utilization of phosphite take place are important, it is certainly beyond the scope of the present study. We have modified the 'Shift in microbial P-utilization around the GOE' section to emphasize these points.

Response to Reviewer #2:

The authors neatly addressed the points the reviewers outlined and made changes to the manuscript which improve its argument for the relationships between phosphite, iron minerals, and the greater early biosphere. While constrained to HFOs, I think this paper opens up space for other experiments to explore behaviors between reactive minerals and phosphite chemistry. This paper is worthy of publication.

Response: We thank the reviewer for this comment and acknowledging the importance of our study.

Response to Reviewer #3:

The authors have successfully answered all of my concerns. I would like to see the publication of the manuscript as soon as possible.

Response: We thank the reviewer for this comment and positive feedback.

References

1. Brady, M. P., Tostevin, R. & Tosca, N. J. Marine phosphate availability and the chemical origins of life on Earth. *Nat. Commun.* **13**, 5162 (2022).
2. Rasmussen, B., Muhling, J. R. & Tosca, N. J. Nanoparticulate apatite and greenalite in oldest, well-preserved hydrothermal vent precipitates. *Sci. Adv.* **10**, eadj4789 (2024).
3. Rasmussen, B., Muhling, J. R., Suvorova, A. & Fischer, W. W. Apatite nanoparticles in 3.46–2.46 Ga iron formations: Evidence for phosphorus-rich hydrothermal plumes on early Earth. *Geology* **49**, 647–651 (2021).
4. Van Mooy, B. A. S. *et al.* Major role of planktonic phosphate reduction in the marine phosphorus redox cycle. *Science (80-)*. **348**, 783–785 (2015).
5. Pech, H. *et al.* Detection of geothermal phosphite using high-performance liquid chromatography. *Environ. Sci. Technol.* **43**, 7671–7675 (2009).
6. Han, C. *et al.* Determination of phosphite in a eutrophic freshwater lake by suppressed conductivity ion chromatography. *Environ. Sci. Technol.* **46**, 10667–10674 (2012).
7. Qiu, H., Geng, J., Ren, H. & Xu, Z. Phosphite flux at the sediment–water interface in northern Lake Taihu. *Sci. Total Environ.* **543**, 67–74 (2016).
8. Pasek, M. A., Sampson, J. M. & Atlas, Z. Redox chemistry in the phosphorus biogeochemical cycle. *Proc. Natl. Acad. Sci.* **111**, 15468–15473 (2014).
9. Martínez, A., Osburne, M. S., Sharma, A. K., DeLong, E. F. & Chisholm, S. W. Phosphite utilization by the marine picocyanobacterium *Prochlorococcus* MIT9301. *Environ. Microbiol.* **14**, 1363–1377 (2012).
10. Boden, J., Som, S. S., Brazelton, W. J., Anderson, R. A. & Stüeken, E. E. Evaluating serpentinization as a source of phosphite to microbial communities in hydrothermal vents. *Geobiology In press*, (2025).
11. Sánchez-Baracaldo, P. Origin of marine planktonic cyanobacteria. *Sci. Rep.* **5**, 17418 (2015).
12. Boden, J. S., Zhong, J., Anderson, R. E. & Stüeken, E. E. Timing the evolution of phosphorus-cycling enzymes through geological time using phylogenomics. *Nat. Commun.* **15**, 3703 (2024).
13. Pasek, M. A., Harnmeijer, J. P., Buick, R., Gull, M. & Atlas, Z. Evidence for reactive reduced phosphorus species in the early Archean ocean. *Proc. Natl. Acad. Sci.* **110**, 10089–10094 (2013).
14. Herschy, B. *et al.* Archean phosphorus liberation induced by iron redox geochemistry. *Nat. Commun.* **9**, 1346 (2018).